# Synthetic recovery of impulse propagation in myocardial infarction via silicon carbide semiconductive nanowires

Paola Lagonegro[1,2,10], Stefano Rossi [3,10], Nicolò Salvarani [4,5], Francesco Paolo Lo Muzio [3,6], Giacomo Rozzi[3,4], Jessica Modica [4,5], Franca Bigi [7,1], Martina Quaretti [7,1], Giancarlo Salviati [1], Silvana Pinelli[3], Rossella Alinovi[3], Daniele Catalucci [4,5], Francesca D'Autilia [4], Ferdinando Gazza[8], Gianluigi Condorelli [4,9], Francesca Rossi [1,11] & Michele Miragoli [3,4,11✉]

Myocardial infarction causes 7.3 million deaths worldwide, mostly for fibrillation that electrically originates from the damaged areas of the left ventricle. Conventional cardiac bypass graft and percutaneous coronary interventions allow reperfusion of the downstream tissue but do not counteract the bioelectrical alteration originated from the infarct area. Genetic, cellular, and tissue engineering therapies are promising avenues but require days/months for permitting proper functional tissue regeneration. Here we engineered biocompatible silicon carbide semiconductive nanowires that synthetically couple, via membrane nanobridge formations, isolated beating cardiomyocytes over distance, restoring physiological cell-cell conductance, thereby permitting the synchronization of bioelectrical activity in otherwise uncoupled cells. Local in-situ multiple injections of nanowires in the left ventricular infarcted regions allow rapid reinstatement of impulse propagation across damaged areas and recover electrogram parameters and conduction velocity. Here we propose this nanomedical intervention as a strategy for reducing ventricular arrhythmia after acute myocardial infarction.

[1] Istituto dei Materiali per l'Elettronica e il Magnetismo (IMEM), National Research Council CNR, Parco Area delle Scienze 37/A, 43124 Parma, IT, Italy. [2] Istituto di Scienze e Tecnologie Chimiche "Giulio Natta", Consiglio Nazionale delle Ricerche (SCITEC-CNR), Via A. Corti 12, 20133 Milan, IT, Italy. [3] CERT, Centro di Eccellenza per la Ricerca Tossicologica, Dipartimento di Medicina e Chirurgia Università di Parma, Via Gramsci 14, 43124 Parma, IT, Italy. [4] Humanitas Research Hospital — IRCCS, Via Manzoni 56, 20089 Rozzano (Milan), IT, Italy. [5] Istituto di Ricerca Genetica Biomedica (IRGB), National Research Council CNR, UOS Milan Via Fantoli 16/15, 20138 Milan, IT, Italy. [6] Dipartimento di Scienze Chirurgiche Odontostomatologiche e Materno-Infantili, Università di Verona, Policlinico G.B. Rossi, - P.le L.A. Scuro 10, 37134 Verona, IT, Italy. [7] Dipartimento di Scienze Chimiche, della Vita e della Sostenibilità Ambientale, Università di Parma, Parco Area delle Scienze, 11/a - 43124, Parma, IT, Italy. [8] Dipartimento di Scienze Medico-Veterinarie, Università di Parma, via del Taglio 10, 43126 Parma, IT, Italy. [9] Department of Biomedical Sciences Humanitas University, Via Rita Levi Montalcini 4, 20090 Pieve Emanuele Milan, IT, Italy. [10] These authors contributed equally: Paola Lagonegro, Stefano Rossi. [11] These authors jointly supervised this work: Francesca Rossi, Michele Miragoli. ✉email: michele.miragoli@unipr.it

The acute phase of myocardial infarction (MI) counts for 12% of patients developing arrhythmias and causes cardiac arrest. Unfortunately, the real number is unknown as many patients are encountered when already dead[1]. Indeed, sudden cardiac death (SCD) following MI is challenging for clinicians as 50% of cases of coronary artery diseases are not recognized clinically and SCD occurs as its first symptom. The STICH (Surgical Treatment of Ischemic Heart Failure) trial showed that the 5-year cumulative incidence of SCD after coronary artery bypass graft (CABG) was 8.5% (the highest monthly rate of SCD occurred in the 31- to 90-day time period)[2], suggesting that CABG patients, with or without improvement of the left ventricular ejection fraction, may remain at risk of SCD[3]. The infarcted area becomes less excitable thus blocking the conduction and favoring the reentrant electrical circuits, including ventricular fibrillation, which often originates from the scarred regions. These arrhythmias can be sustained mainly by the zig-zagging slow conduction[4] and the electrical remodeling[5]. Apart from conventional pharmacological therapies[6], many investigators attempt to reduce MI-arrhythmogenesis by stem cell[7], genomic[8], epigenomic[9], or tissue engineering[10] treatments. Unfortunately, those interventions do not act rapidly[11,12]. More recently, nanomedicine and nanotechnology have been proposed for the treatment of cardiovascular disease. Although cardiac nanomedicine is still in its infancy, the present and other authors have provided evidence of better recovery from heart failure[13], ischemic heart disease[14], and atherosclerosis-induced MI[15]. Conductive epicardial patches enriched with single-walled carbon nanotubes (SWCNT) and conductive ink have been recently shown to recover impulse propagation in canine-damaged hearts[16]. Despite a dispute on SWCNT toxicity[17], the mechanisms underlying the anti-arrhythmic effect and the ability to synchronize cardiac tissue over the distance of the patches remain to be understood. Indeed, if the "synthetic" conductivity cannot be controlled, it may result in slowing/quickening of conduction and thus create a pro-arrhythmic environment itself. Bioresorbable chitosan patches enriched with graphene-oxide gold nanosheets have also been proposed for restoring impulse conduction[18]. This interesting intervention aimed for the re-expression of Connexin-43 in the infarct/peri-infarct zone and required 5 weeks or more for a two-fold increase in impulse propagation, so exposed the heart to early phases of MI-induced arrhythmias. Nevertheless, nanomaterials capable of rapidly restoring impulse propagation, with the reinstatement of "physiological cardiac conductivity" are highly desirable. Silicon carbide nanowires (SiC-NWs) have been recognized as a possible candidate for the development of a new generation of implantable nano-devices since they are chemically inert, semiconducting[19] and compatible with the biological environment[20]. In this work, we demonstrate that injectable and biophysically inspired SiC-NWs are capable of electrotonically synchronizing isolated cardiomyocytes via the formation of membrane nanobridges (MNBs)[21]. They biophysically regenerate physiological electrical conductance and synthetically restore electrograms (EGs) parameters and impulse propagation in left ventricular MI regions within 5 h following their injection.

## Results

### SiC-NWs electrically couple cardiomyocytes over a distance.
HL1 is an atrial cell line that replicates in culture and has spontaneous electrical activity and impulse propagation[22]. We assessed whether crystalline cubic (3 C) SiC-NWs of ca. 20 nm width and ca. ten micrometers long (Fig. 1a) with a ζ-potential of −10 mV, were biocompatible with those cells. We did not observe changes in cell viability or intracellular ATP levels 24 and 48 h after exposure to NWs at concentrations of 15 µg/ml and 50 µg/ml (Fig. 1b, c), similar to what we have previously described in other cell lines[20]. Moreover, SiC-NWs were not completely internalized by cells and did not modulate ROS production or induce lipid peroxidation (Fig. 1d-e, Supplementary Fig. 1) at either concentration. Furthermore, SiC-NWs did not interfere with the cell cycle (Fig. 1f-g).

SiC-NWs were immediately covered by the cell membrane, resulting in a nanoscale membrane protrusion capable of merging with the neighboring cells (Fig. 2a). Of note, double patch-clamp clamping revealed that SiC-NWs-produced MNBs electrically synchronized two cells over distance with an average conductance of $3.98 \pm 0.97$ nS; by contrast natural cell–cell coupling resulted in a junctional current of $14.13 \pm 2.41$ nS (Fig. 2b, c). This discrepancy was caused by the more heterogeneous lengths of the cell–cell contacts in the natural ($9.36 \pm 6.88$ µm) vs. synthetic ($4.70 \pm 0.65$ µm) condition. We obtained similar MNBs generation and thus the fusion with the distance cell by SiO$_2$-NWs (Supplementary Fig. 2), but there was no junctional current (Supplementary Fig. 2c), a finding suggesting that only semi-conductive SiC-NWs and not insulated SiO$_2$-NWs are capable to support junctional current over distance.

### SiC-NWs synthetically connect cardiac cells and synchronize electrical activity over a distance.
We then investigated whether the generation of the MNB was directly orchestrated by SiC-NWs. Dual-beam FIB/SEM microscopy revealed that HL1 cells form MNBs only in the presence of NWs in their media and, by doing so, physically connect two distant cells (Fig. 3a). Thanks to the FIB sectioning we were able to dissect MNBs along desired planes, with SEM allowing us to visualize orthogonal cross-sections and verify the presence of SiC-NWs within them (Fig. 3a, blue arrowheads).

At 4 h, SiC-NWs were interacting with the outer part of the cellular membrane, becoming enveloped within it, and eliciting the formation of MNBs. Of note, SiC-NWs were several micrometers long and tended easily to form bundles and partially overlap to cover a very long distance. Therefore, depending on the number of SiC-NWs deposited, more than a single SiC-NW can be covered by the same MNB, resulting in the capability to connect more cells creating a synthetic network of cell activity in vitro. Live-confocal imaging revealed that MNB formation was orchestrated by actin filaments that linearly colocalized with the SiC-NWs by ~30% after 30 min in HL1 cells (Fig. 3b,c, Supplementary Video 1) and in cardiac myofibroblasts (Supplementary Fig. 3, Supplementary Video 2).

HL1 cells generated spontaneous action potentials (APs) that became synchronized between a "source" cell (where the AP initiated) and a "sink" cell (where the AP was propagated) (Fig. 4). While the quality of the AP duration and the upstroke velocity ($dV/dt_{max}$) of the sink cells was similar to that of the source cells and control cells (without SiC-NWs), this was not the case for the AP amplitude, which was significantly reduced in the sink cells vs. to control (control=$14.8 \pm 0.9$ dF/F, Cell Source = $13.3 \pm 0.5$ dF/F, Cell Sink = $11.9 \pm 0.6$ dF/F, Fig. 4b). We observed synchronized electrical activity up to a maximum cell–cell distance of 360 µm (Fig. 4c), similarly to what we observed in the past for the electrotonic propagation across passive electrically coupled myofibroblasts and Cx43 transfected HeLa cells[23,24]. Likewise, the intracellular Ca$^{2+}$ transient was also synchronized between distant cells linked by SiC-NWs (Fig. 5, Supplementary Video 3). Indeed, intracellular Ca$^{2+}$ amplitude was statistically different in the source and sink cells (Fig. 5c and d); moreover, there was a direct correlation between the initiation of intracellular Ca$^{2+}$ delay and the cell–cell distance ($R^2 = 0.93$).

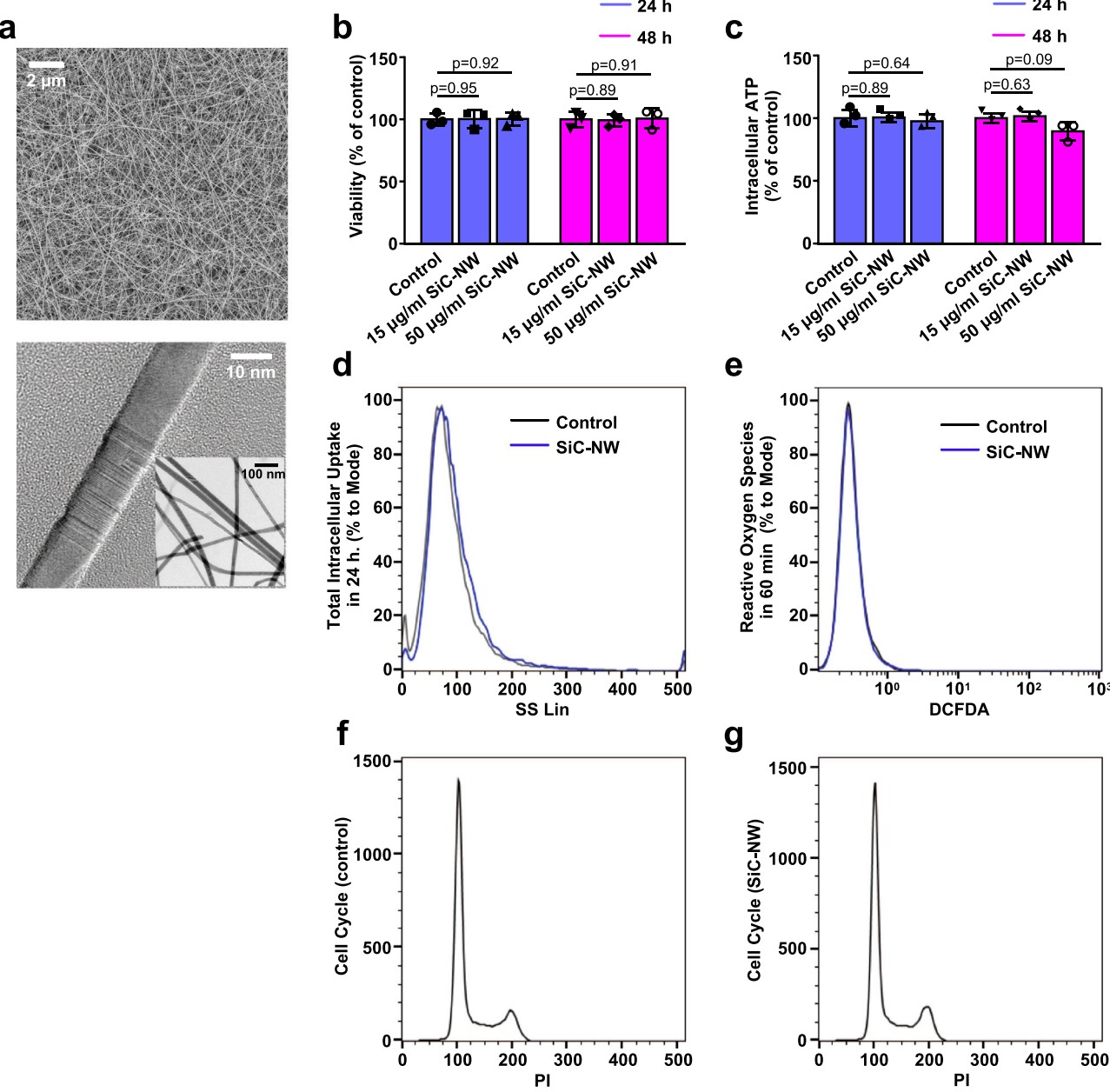

**Fig. 1 Growth of silicon carbide nanowires (SiC-NWs): physicochemical and biocompatibility characterizations. a** Representative bright-field SEM (top) and TEM (bottom) images of SiC-NWs detached from the substrate ($n = 50$ experiments repeated with similar results). **b** Viability of HL1 cardiomyocytes at 24 h (blue) and 48 h (pink) in control (no SiC-NWs) and SiC-NWs conditions ($n = 3$ biologically independent experiments in each group). Data are represented as mean ± SD. Unpaired two-sided Student's $t$-test (statistical significance set at $p < 0.05$). Source data are provided as a Source Data file. **c** Same as **b** for intracellular ATP concentration. **d** Total intracellular uptake of SiC-NWs by HL1 cells ($n = 3$ experiments repeated with similar results). SS Lin Side scatter on a linear scale. **e** 60 min of reactive oxygen species (ROS) production from HL1 cells cultured w or w/o 50 μg/ml of SiC-NWs ($n = 3$ experiments repeated with similar results). DCFDA dichlorodihydrofluorescein diacetate. **f** Cell cycle characteristics of control (no SiC-NWs) HL1 cells ($n = 3$ experiments repeated with similar results). PI propidium iodide. **g** Same as **f** after 48 h of exposure to 50 μg/ml SiC-NWs.

The maximum cell–cell distance capable of keeping a synchronized Ca$^{2+}$ activity was 358 μm, confirming that SiC-NWs support passive electrotonic propagation in a length-dependent manner.

**SiC-NWs re-established excitability and refractoriness in myocardial infarction.** Having demonstrated that biophysically inspired SiC-NWs can synchronize the cellular electrical activity over a distance in vitro via the creation of MNBs, we next investigated the possibility of acutely recovering impulse propagation in a 2 × 2 mm cryoinjury area (Cryo)[25,26] of the rat left ventricle, used as a model of acute MI (Fig. 6). The protocol(Fig. 6a) included electromechanical measurement of epicardial activity in situ during (i) the normal beating hearts, (ii) in the same hearts after transmural MI obtained Cryo, (Fig. 6b), and (iii) after triple injections into the MI area of saline solution (Vehicle) containing 1 mg of SiC-NWs (Fig. 6c) forming spontaneous SiC-NWs network into the desired region (Fig. 6d). The vehicle/SiC-NWs injections were carried out at 2 h from cryoinjury because at this time the electrophysiological parameters became stabilized (Supplementary Fig. 4); indeed, QRS

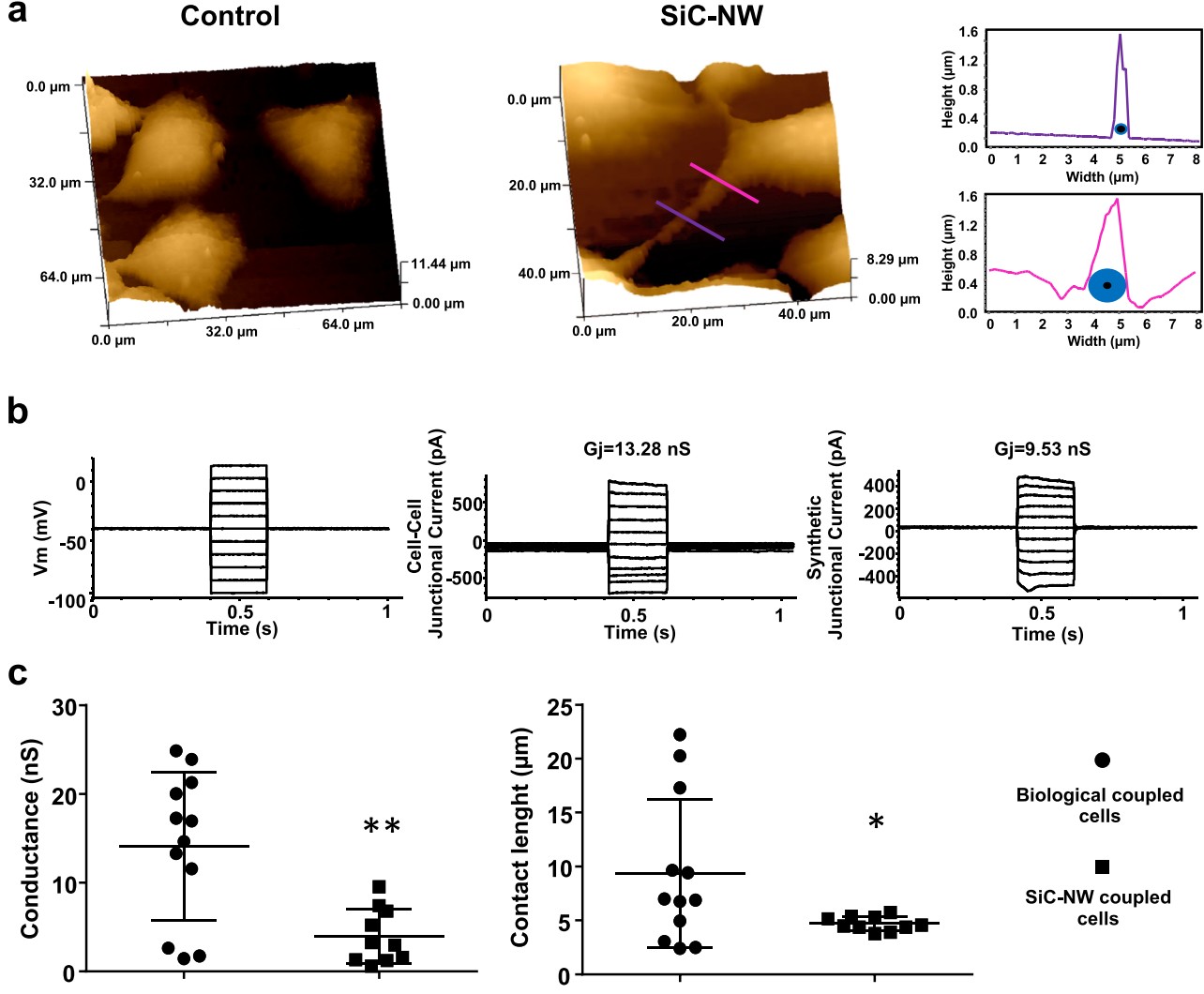

**Fig. 2 Synthetic cell–cell communication via coupling with SiC-NWs. a** Topographical imaging of HL1 cells with Hopping probe scanning ion conductance microscopy (HPICM): biological cell–cell coupling (Control, left); synthetic cell–cell MNB-based coupling obtained with SiC-NWs (SiC-NW, middle); quantification at the purple and pink lines of the middle image (right) ($n = 5$ experiments repeated with similar results). **b** Representative dual patch-clamp traces from biologically and synthetically coupled HL1 cell pairs. Voltage gradient applied to the cell source (left); and junctional conductance in biologically connected (middle) and SiC-NW-connected (right) cell pairs ($n = 5$ experiments repeated with similar results). **c** Cell–cell conductance (left) and contact length (right) for biologically coupled and synthetic SiC-NW-coupled HL1 cell pairs. $n = 12$ for biological coupled cells, black circles; $n = 10$ for SiC-NW coupled cells, black squares. Data are represented as mean ± SD. Unpaired two-sided Student's *t*-test. *$p = 0.046$ vs. biological coupled cells; **$p = 0.0017$ vs. biological coupled cells. Source data are provided as a Source Data file.

amplitude—an indicator of damaged tissue—decreased immediately after cryoinjury, and stabilized after 2 h. As expected, the cryoinjury invariable affected excitability of the damaged/necrotic tissue (Fig. 7a). The activation time (AT) significantly increase from $12.12 \pm 2.62$ to $14.04 \pm 3.75$ ms, as were the QRS duration (from $18.11 \pm 3.29$ to $19.68 \pm 3.83$ ms) and the RR interval (from $567.35 \pm 138.62$ to $588.48 \pm 189.2$ ms), whereas RT interval decreased (from $19.5 \pm 4.22$ to $15.95 \pm 3.60$ ms). Excepting the QRS complex duration, the other parameters were recovered toward physiological values after 5 h following SiC-NWs injection (AT $= 11.17 \pm 2.7$ ms, QRS $= 23.14 \pm 7.13$ ms, RR $= 489.51 \pm 140.82$ ms, RT $= 20.1 \pm 6.01$ ms).

**SiC-NWs re-establish impulse propagation.** We then decided to investigate the homogeneity/inhomogeneity of cardiac conduction in the MI region 5 h after SiC-NWs injection. To this end, we injected a cathodal current from selected electrodes to acquire

isochrones maps in the following condition: (i) after opening the chests, (ii) 2 h after cryoinjury, (iii) 5 h after injection with vehicle or SiC-NWs. Central unipolar cathodal stimulation in the normal myocardium was characterized by an elliptic propagation with the major axis oriented parallel to the epicardial fiber direction (Fig. 7b). After myocardial cryoinjury, the normal propagation disappeared especially in the region surrounding the damaged area. Injection of the vehicle was not associated with re-established spatiotemporal distribution, wherein conduction blocks occurred. However, a physiological scenario was reinstated 5 h following SiC-NWs injection. Spatiotemporal phase-map analysis of the local conduction velocities[27] measured around the stimulation electrodes as depicted in Fig. 7c revealed a similar nondisperse distribution of data before cryoinjury and 5 h after SiC-NWs injection whereas this was not the case for the untreated infarcted hearts.

Analysis of the electrogram parameters (Fig. 8a-g) showed that total AT already displayed a statistically significant difference at the

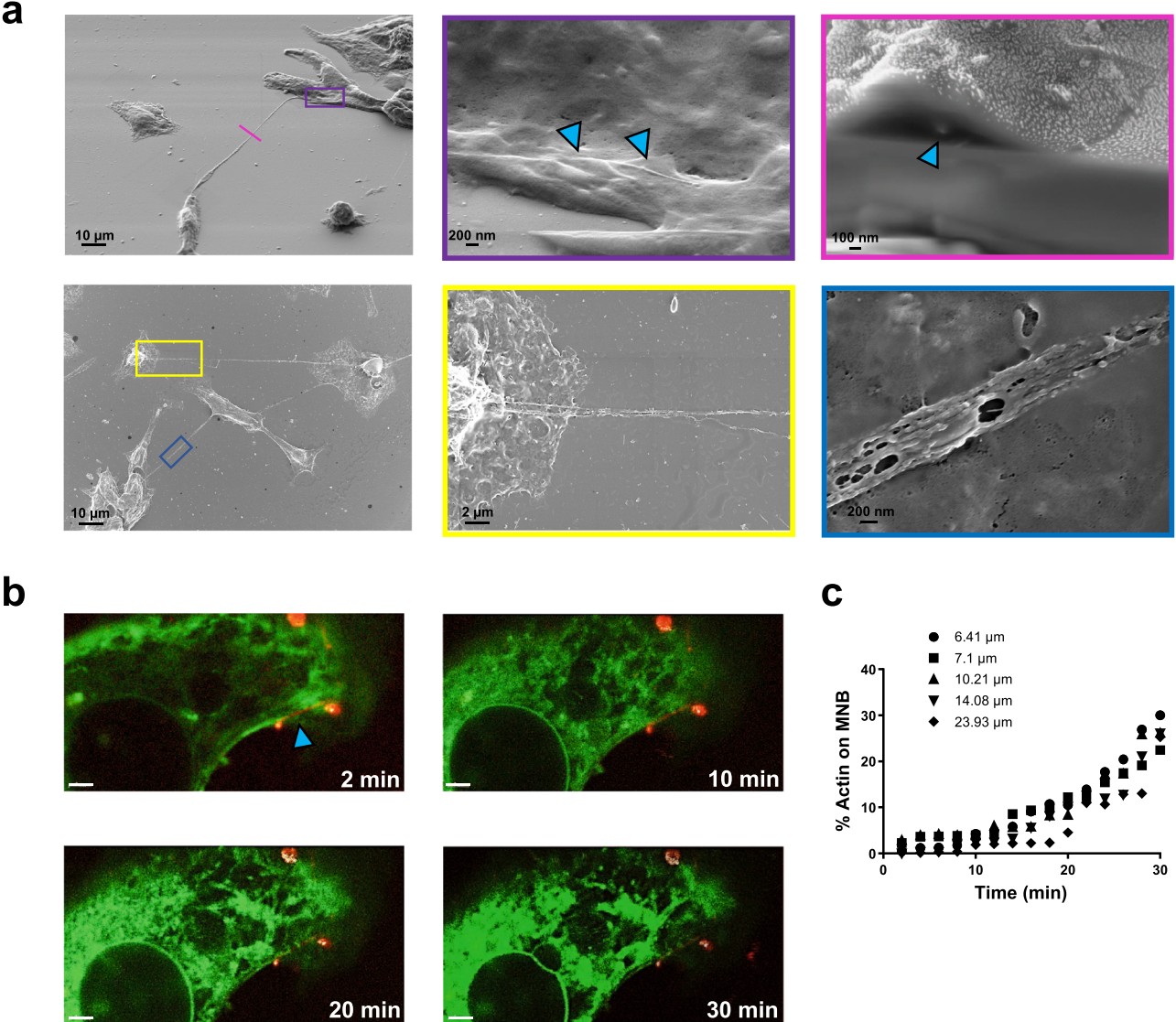

**Fig. 3 Interaction of SiC-NWs with membranes in HL1 cells. a** Top Left. SEM image of NWs forming an MNB physically coupling two distant cells. Top Middle. A blow-up of the purple box in the left image, showing partial internalization of SiC-NWs (blue arrowheads). Top Right. A focused Ion Beam was used to cut the MNB at the pink line in the left image (blue arrowhead points to SiC-NW). Bottom Left. Same as **a**, showing multicellular synthetic coupling created by SiC-NWs. Bottom Middle. A blow-up of the yellow box in the left image, showing that several SiC-NWs can form branches connecting multiple cells. Bottom Right. A blow-up of the blue rectangle, showing that several NWs can form a single MNB ($n = 5$ experiments repeated with similar results). **b** Time-lapse recording of the biodynamic interface an HL1 cell live-stained for actin (green) and SiC-NW (red). Images were acquired every 2 min for a 30 min period; Scale bars: 2 μm. Blue arrowhead indicates the position of the SiC-NW ($n = 17$ biological independent cells). **c** MNB formation triggered by actin for SiC-NWs partial internalization ($n = 5$). Legend numbers indicate SiC-NW lengths. Source data are provided as a Source Data file.

3rd hour after the SiC-NWs injection, alongside abolishment of prolongation of $QT_c$ ($R^2$: Cryo 0.78, Cryo+Vehicle 0.95, Cryo+ SiC-NWs 0.48) and RT interval ($R^2$: Cryo 0.93, Cryo+Vehicle 0.53, Cryo+SiC-NWs 0.61) but no for QRS complex duration ($R^2$: Cryo 0.64, Cryo+Vehicle 0.06, Cryo+SiC-NWs 0.72, Fig. 8a-d). Of note, the injection of SiC-NWs abruptly ended the increment in T-wave duration ($R^2$: Cryo 0.72, Cryo+Vehicle 0.77, Cryo+SiC-NWs 0.04 (Fig. 8e). Moreover, the RR interval was linearly reduced over time after injection of either the Vehicle alone or with SiC-NWs ($R^2$: Cryo 0.02, Cryo+Vehicle 0.99, Cryo+SiC-NWs 0.98) (Fig. 8f). However, we did not observe changes in the cardiac kinematics, as expected by the limited damaged area on the left ventricle (Supplementary Fig. 5) although, a nonsignificant trend of recovery was observed at the perimeter and the area of the deformed tissue during the cardiac cycle as an index of ameliorating ventricular compliance (Supplementary Fig. 6).

## Discussion

Here we provided the experimental evidence that biocompatible SiC-NWs synthetically couple cardiomyocytes over distance, allowing for the recovery of impulse propagation in rat's MI model. The opportunity to employ silicon-based NWs materials for triggering, measuring, and monitoring AP propagation originated mainly from neurophysiology[28], where silicon NWs can be functionalized and adapted for photoelectrochemical modulation[29], the mapping neuronal circuits[30] or for creating "cyborg tissues" for pharmacological screening or future brain implants[31]. Silicon NWs have been recently proposed to create injectable cell-silicon hybrids actable via photo-stimulation (mainly myofibroblasts, the cells that populated a myocardial infarction) for synchronizing heterocellular electrical activity in the infarcted heart[32]. These interesting studies coupled insulating ($SiO_2$) NWs with an unexcitable cell (the myofibroblast) which

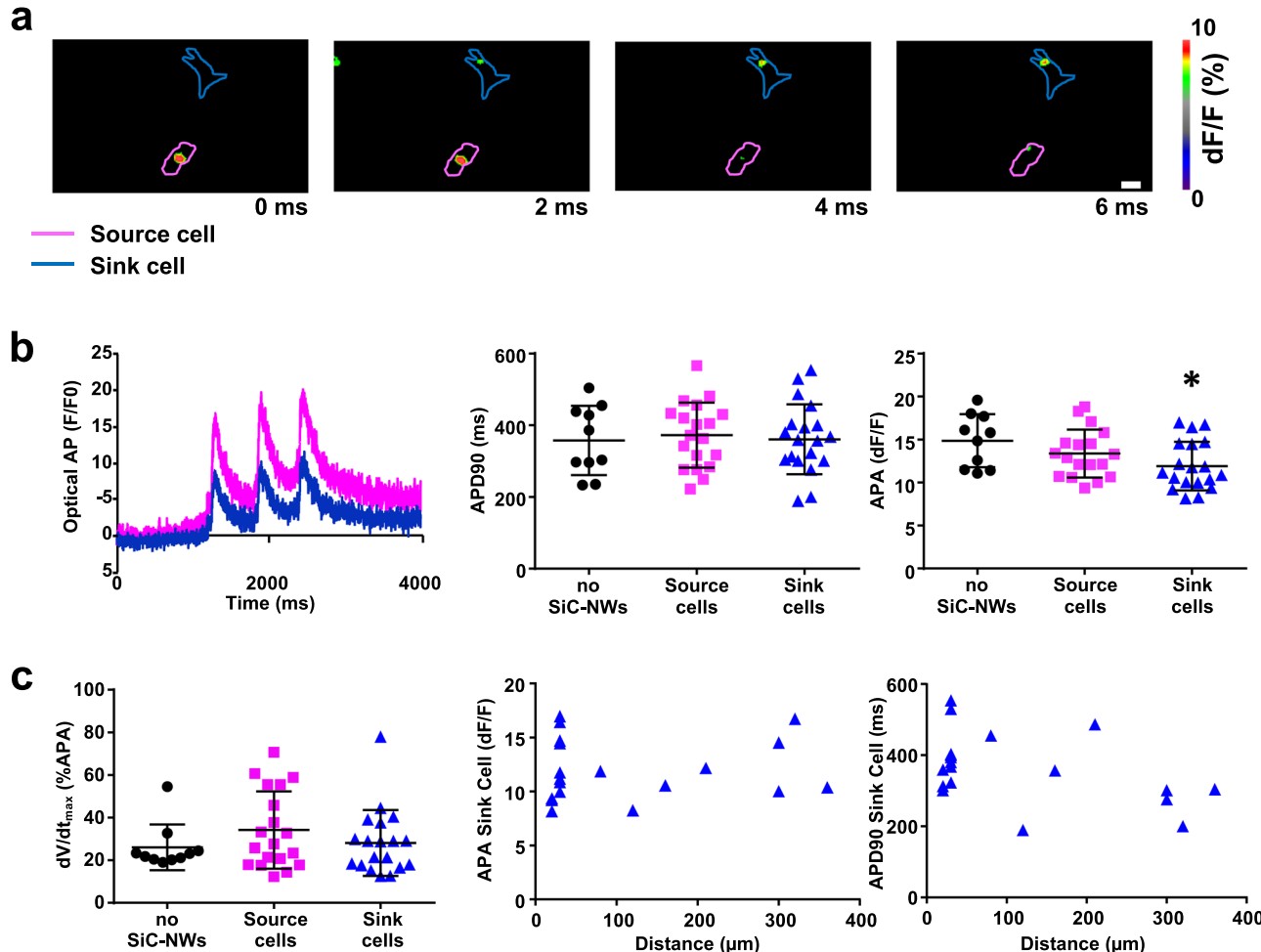

**Fig. 4 Electrotonic optical action potential synchronization over distance in synthetically coupled cells. a** Color-coded images of an optical action potential spontaneously initiated in the pink HL1 cell (source) and rapidly propagating via SiC-NWs, to the blue HL1 cell (sink); ×20 magnification, scale bar = 10 μm. $n = 19$ experiments repeated with similar results. **b** Left Panel. Representative spontaneous APs propagated from a source cell (pink) to a sink cell (blue). $n = 19$ experiments repeated with similar results. Middle Panel. Optical AP duration calculated at 90% of repolarization in control HL1 cells (no SiC-NWs, $n = 10$, black dots), source cells ($n = 19$, pink squares), and sink cells ($n = 19$, blue triangles). Right Panel. Same as Middle Panel for AP amplitude. **c** Left Panel. Maximal upstroke velocity ($dV/dt_{max}$) calculated in controls ($n = 10$, black dots), sources ($n = 19$, pink squares) and sinks ($n = 19$, blue triangles). Middle Panel. AP amplitude values ($n = 19$, blue triangles) for cell sinks vs. cell–cell distance. Right Panel. Action Potential duration values ($n = 19$, blue triangles) for cell sink vs. cell–cell distance. Data are expressed as mean ± SD for the dot plots graphs, real values for the blue triangles graphs. Unpaired two-sided Student's t-test. Statistical significance set at $p < 0.05$. *$p = 0.01$ vs no SiC-NWs. Source data are provided as a Source Data file.

we observed to create gap junctions with surrounding cardiomyocytes in vitro[33] and supporting impulse propagation via the supernormal conduction[24,34]. In the current study, we grew SiC-NWs that do not require cell grafting for the support of impulse propagation but, once cultured with cardiomyocytes or myofibroblasts, can create actin-triggered MNB networks that synthetically transfer the electrotonic potential from cell-to-cell over distance. SiC-NWs were not completely internalized by the cells[35] and, probably due to their structural properties, neither increase ROS production[36] nor affect the cell cycle[37]. It has been recently observed that similar membrane formation allows heterocellular coupling of cardiac and noncardiac cells in the scar tissue and acts as a possible substrate of heterocellular coupling[38], and the transfer of mitochondria between stem cells and cardiomyocytes[39]. In our experiments, we compared the MNB formation as triggered by SiO2- and SiC-NWs; only semiconducting SiC-NWs supported coupling, because of their conductivity (i.e., ~11.4 μS cm$^{-1}$)[19], and therefore impulse propagation over distances. However, SiC-NWs coupled cells had

reduced junctional currents compared to physically coupled HL1 cardiomyocytes[40], but this reduction does not impede the correct AP transfer via MNB formation.

Weingart et al. and Wilder et al. showed in in vitro coupled cardiomyocytes and mathematical simulations that AP transfer is possible for electrical conductance values lower than 3.98 nS[41,42]. Nonetheless, the junctional current not only depends on the degree of coupling but also on the number of gap junctions present[33]. This is not the case when HL1 cells were synthetically coupled by SiC-NWs, wherein the contact lengths depended on MNB edge dimensions (cf. Fig. 2c). Such coupling allowed sustainable AP propagation (albeit reducing AP amplitude in sink cells, possibly on account of electrotonic impulse propagation delays across MNBs[23]) and intracellular Ca$^{2+}$ transient synchronization in the cell network.

We thus provided preclinical level evidence that injection of SiC-NWs into the injured cardiac tissue restores conduction blocks deriving from the damaged portions of the ventricles. Isochrones maps revealed resolution of conduction block and the

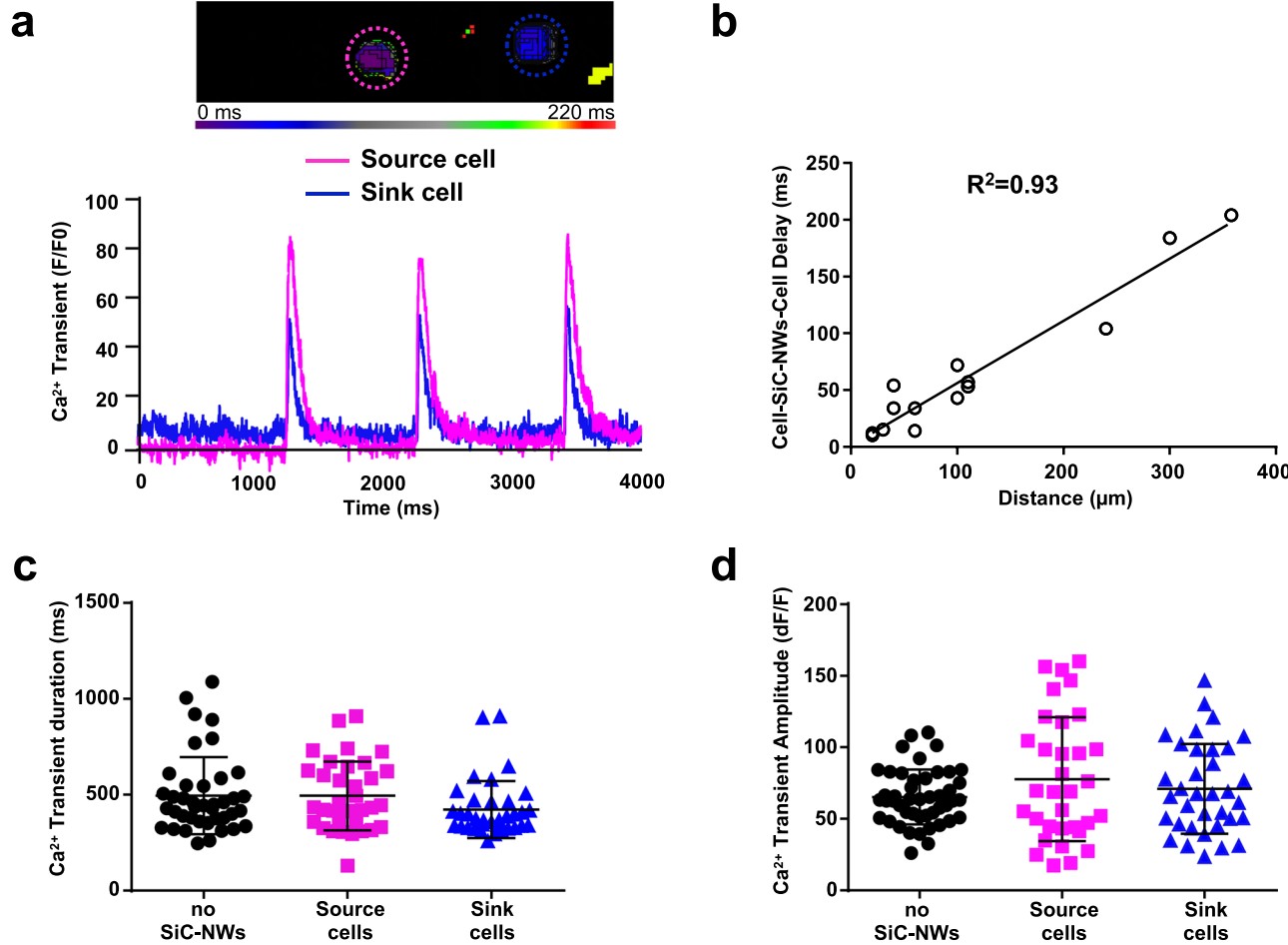

**Fig. 5 Synchronization of electrotonic intracellular Ca²⁺ transients over distance in synthetic cell–cell coupling. a** Top. Color-coded time course of Ca²⁺ initiation in two MNB-connected HL1 cells. Pink-dashed circle, source cell; blue-dashed circle, sink cell. Bottom. Relative series of spontaneous intracellular Ca²⁺ transient traces from the source (pink) and sink (blue) cell. $n = 35$ experiments repeated with similar results. **b** Intracellular Ca²⁺ transient delay from coupled cell pairs connected by SiC-NWs over distance, showing a linear correlation ($n = 15$). **c** Intracellular Ca²⁺ transient duration in control (no SiC-NWs, $n = 48$, black dots) source cells ($n = 35$, pink squares), and sink cells ($n = 35$, blue triangles). **d** Same as **c**, but for Ca²⁺ amplitude. Data expressed as mean ± SD. Unpaired two-sided Student's $t$-test (significance set at $p < 0.05$). Source data are provided as a Source Data file.

reduction in conduction dispersion 5 h following SiC-NWs treatment, indicating that in the damaged tissue the propagation was uniformly reinstated to a level observed in the untreated, normal hearts. The prolongations of AT and QTc are well-known indexes of acute myocardial ischemia/infarction[43] and substrates for ventricular fibrillation in both idiopathic and inherited arrhythmias[44–46]. We found that SiC-NWs restored AT, RT, and RR intervals towards the physiological levels. Of note, the QTc prolongation and the RT interval decrement were also abolished over time, even if the QRS interval remained prolonged in all three conditions. Moreover, there was a stronger effect on T-wave duration (cf. Fig. 8e); indeed, the treatment immediately abolished the linear increment in the T-wave observed in the Cryo and Cryo + Vehicle groups, suggesting an interesting responsivity of SiC-NWs also in the repolarization phase of the EGs, and that it has a primary role in the preservation of QTc and RT parameters. Regarding the RR interval, there was a significant increment in the average value in the Cryo group with a reduction toward a physiological level following SiC-NWs treatment. Finally, while heart rate was preserved over time in the Cryo group the Vehicle and treated groups display a linear decrease over time because saline injections in the cardiac tissue may affect the heart rate as reported elsewhere[47].

We have shed light on the capacity of biophysically inspired SiC-NWs to create actin-based MNB networks, mimicking the physiological cell–cell cardiac communication. The nanowires are partially and rapidly internalized by the cardiomyocytes and synthetically transfer electrotonic impulse propagation over distance, by keeping the physiological grade of cardiac conductance. In MI, the treatment with SiC-NWs rapidly relieved possible conduction blocks, thereby reestablishing EGs parameters and the inhomogeneity in conduction. This intervention may support CABG in reinstating bioelectrical activity in the damaged myocardium.

We decided to perform local ventricular cryoinjury instead of coronary ligation for mimicking MI[48]. This was done to standardize the infarcted region areas and avoid the presence of late-recruited cardiac myofibroblasts which may act as confounders because capable of electrotonically supporting, impulse propagation[24,38]. Notwithstanding the acute response to SiC-NWs and the absence of ROS and TBARs production, a long-term toxicological evaluation will be mandatory to assess the possibility of adopting SiC-NWs for an anti-arrhythmic nano-medical therapy after CABG. Further investigations are also necessary to monitor long-term grafting of SiC-NWs into the MI area and to physically orientate injectable SiC-NWs along the

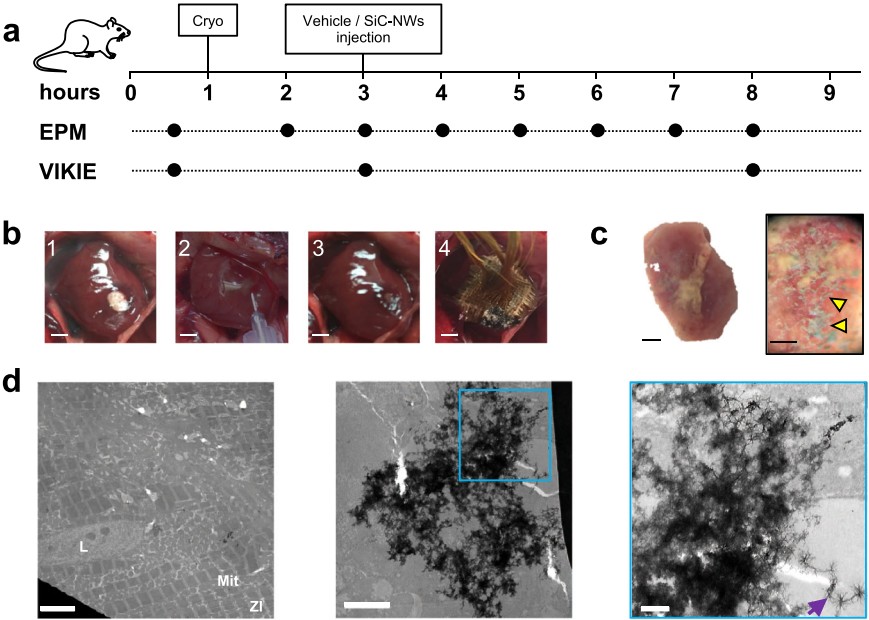

**Fig. 6 Electrophysiological profile of recovered sinus propagation in a cryoinjured portion of the heart via SiC-NWs myocardial injection. a** Timeline of the experimental adopted protocol for providing in situ left ventricular cryoinjury (Cryo) and Vehicle/SiC-NWs injection. EPM: epicardial potential mapping recording. ViKiE Video Kinematic Evaluation recording. **b** Photographs of the in situ heart for the cryoinjury (1), SiC-NWs injection (2), injected area (3), and epicardial grid (4) covering the injured and the surrounding portion of the ventricle ($n = 10$ experiments repeated with similar results); Scale bars: 2 mm. **c** Stereomicroscopy images of the transmural cryoinjured area (pale region left) containing 1 mg of SiC-NWs (yellow arrowheads, right); Scale bars: 500 μm (left), 80 μm (right). **d** TEM images of the infarcted cardiac tissue with SiC-NWs formed network. Left: macroscopic electron-dense images showing disorganized Z-lines (Zl), mitochondria (Mit), and capillary lumen (L). Middle and right images: area covered by the SiC-NWs network (zoomed in the blue area, purple arrow). $n = 20$ experiments repeated with similar results. Scale bars: 5 μm (left), 2 μm (middle), and 500 nm (right).

epicardial fiber direction to improve the quality of propagation and reduce vulnerability to MI-induced arrhythmias.

## Methods

For cell culture experiments, ethics permission was obtained from the Italian Ministry of Health (Protocol N. D2326/2019), and all isolations followed the directives of European law 63/2010 and Italian law 26/2014 for experimental animal use.

All animal experiments performed were done in accordance with experimental protocols reviewed and approved by the Italian Ministry of Health (approved protocols: PMS53/2009, 281/2017, 989/2017), under strict compliance to the Italian (D.L.4/3/2014) and European (2010/63/UE) guidelines for ethical use of animal models in biological research.

**Silicon carbide and silicon oxide nanowires growth**. Bare 3C-SiC NWs were obtained by wet chemical etching of the SiC/SiOx NWs as reported elsewhere[49]. Briefly, core-shell SiC/SiOx (1.7 < x < 2) NWs were grown on silicon substrates in a Chemical Vapor deposition open tube setup at atmospheric pressure, through a vapor-liquid-solid process catalyzed by iron. The growth was performed on substrates wetted by ferric nitrate in an ethanol solution, flowing CO in $N_2$ carrier (0.4%) as gaseous precursor at 1100 °C for 30 min[37]. The as-grown core/shell structure was made by a crystalline 3C-SiC core with an average diameter of 20 nm, wrapped by an amorphous $SiO_x$ shell giving an average total NW diameter of 60 nm[50]. To study the NW performance as biocompatible conductive nanomaterials, the insulating shell was removed, leaving the bare SiC semiconductor core. It was performed by chemical etching using a standard RCA clean followed by a second chemical treatment with Piranha solution and final etching ($H_2O$:HCl 1:2 and $H_2O$:HF 50:1)[49]. Surface analysis performed by photoelectron spectroscopy[51] showed a slight carbon excess in SiC stoichiometry (Si:C = 0.85 ± 0.05) and a terminating layer (~1 nm thick) compatible with silicon oxycarbide. SiC-NWs and $SiO_2$-NWs have been removed from silicon substrate previous usage by sonication in an ultrasonic bath for 30 min. For the confocal acquisitions, SiC-NWs have been functionalized with amino groups by reacting with 3-aminopropyltriethoxysilane (APTES) (2 μL of a solution 5 mmol/L in toluene) in refluxing toluene. After washing with toluene and ethanol, the SiC-NWs were detached using a sonicating bath. The NWs were collected from ethanol by ultracentrifugation at 15,339 × g at 4 °C for 30 min and dried in the air. The NWs (1 mg) were dispersed in DMSO (1 ml) and 1 μL of DIPEA was added under magnetic stirring. Then, 200 μL of an NHS-Abberior STAR 580 solution in DMSO (1 mg/ml) was added. The reaction was stirred for 2 h at room temperature to allow the formation of the amide bond.

Centrifugation at 15,339 × g at 4 °C, followed by three washing with DMSO, allowed to recover the NWs conjugated with the dye.

**Cell culture**. HL1 mouse atrial cardiomyocytes[52] (Merck, U.S.A., Code: SCC065) were cultured in Claycomb medium supplemented with 10% fetal bovine serum (FBS), 4 mmol/L L-glutamine, 100 U/ml Penicillin, 100 mg/ml Streptomycin, 0.3 mmol/L Ascorbic Acid, and 10 mmol/L Norepinephrine (all from Sigma–Aldrich, USA) as previously described[53]. Cells were plated onto gelatin/fibronectin-coated 22 mm coverslips at the density of 2000 cells/cm². After 24 h in culture, the cells received 50 μg/ml SiC-NWs or $SiO_2$-NWs and all experiments have been performed after 4–5 h from NWs administration. Neonatal rat ventricular myofibroblasts have been isolated from 1–2 day-old ventricles via trypsin/pancreatin digestions and seeded on 22 mm glass coverslips[24].

**Toxicological assays**. We used recently described methods, to show cytocompatibility of SiC-NWs in cancer cells[37] and fibroblasts cell lines[20]. Briefly, we studied cell viability via MTT assay and assessed intracellular ATP levels using the CellTiter-Glo® luminescent cell viability assay (Promega, Madison, WI, USA), according to the manufacturer's recommendations. Flow cytometry, using an FC500™ flow cytometer (Instrumentation Laboratory, MA, USA), was used to evaluate the nanomaterial internalization, the cell cycle progression, and the ROS production. Data were processed with the FlowJo software package (Tree Star Inc., OR, USA). Side-scatter counts (SSC) were acquired to quantify uptake of NWs. The cell phase distribution was determined by DNA content. Briefly, cells were fixed in ethanol, stained with propidium iodide, and then sorted in cytofluorimeter. Intracellular ROS generation was investigated employing 2′,7′-dichlorodihydro-fluorescein diacetate (DCFH-DA). Hydrogen peroxide (50 μmol/L) was adopted as a positive control. The well-established method 'Thiobarbituric Acid Reactive Substances (TBARS) method was used to evaluate lipid peroxidation, using a Cary Eclipse fluorescence spectrophotometer (Varian, Inc., Palo Alto, CA, USA) (excitation 515 nm, emission 545 nm). Values were normalized for the protein concentrations and expressed as a percentage of control.

**SiC-NWs interaction between isolated cardiomyocytes**. HPICM setup and its operation in hopping mode[54] consist of a piezo-controller (ICnano Scanner Controller, Ionscope Ltd, UK) controlled the xyz piezo three-axis translation stage (Physik Instruments, DE) with 100 μm closed-loop travel range in x, y, and 50 μm z directions. The piezo-stage was driven by a high-voltage amplifier (Physik Instruments, DE) connected to the Icnano scanner controller and the pipette electrode head-stage was connected to Multiclamp 700B (Molecular Devices, Crisel

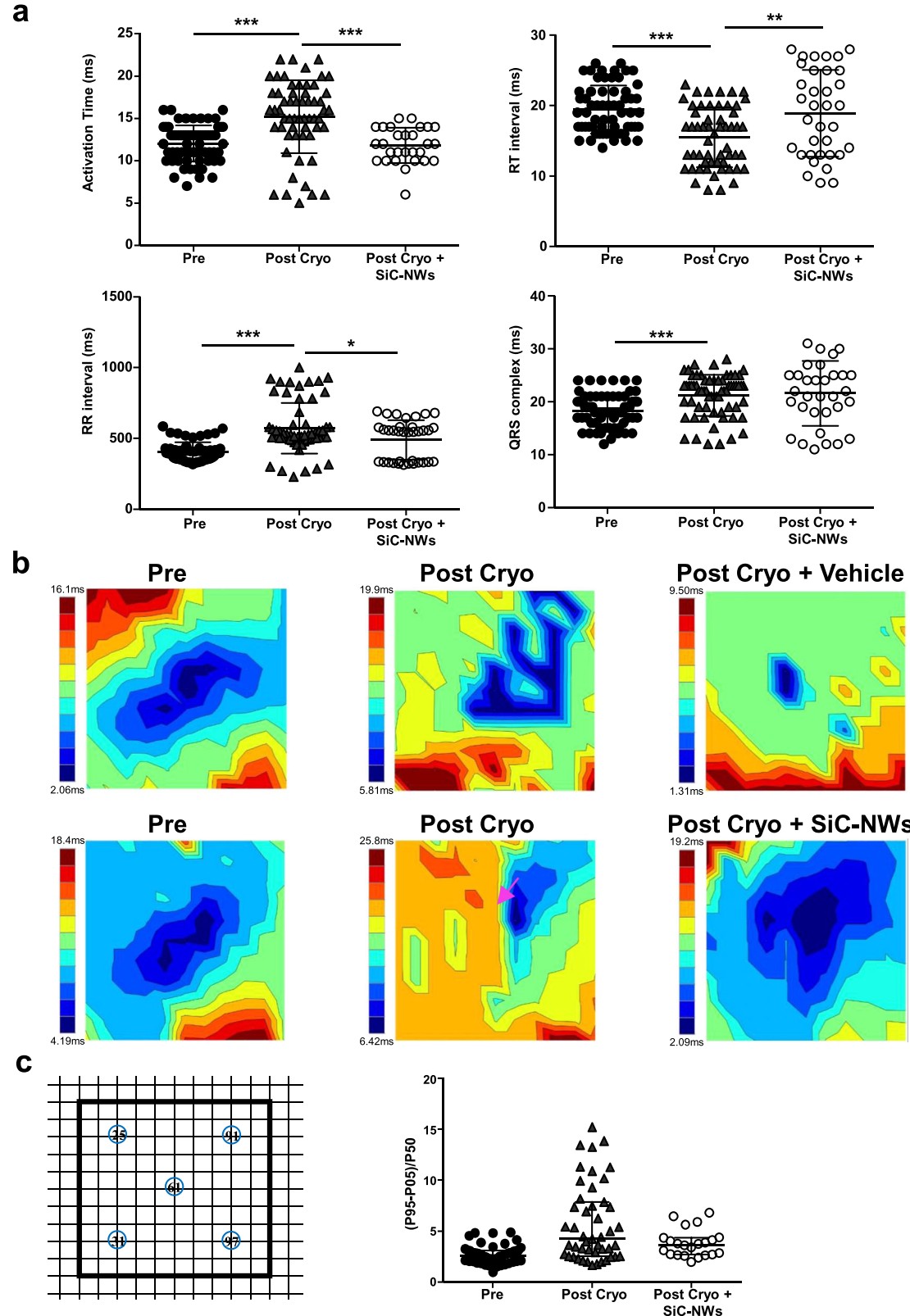

Instruments, IT). The scan head was placed onto an electrical micromanipulator (Scientifica, UK) based on a motorized platform (Scientifica, UK). Preparations were imaged by a Nikon Tei inverted microscope (Nikon Corporation, Japan). Nanopipettes (60 MΩ resistance) were pulled from borosilicate glass with O.D. 1.0 mm and I.D. 0.58 mm (Crisel Instruments, IT) using a laser puller P-2000 (Sutter Inc., USA). Nanopipettes were filled with HBSS (Euroclone, IT). Surface topographical images of the two cells (max 60 × 60 μm, 512 × 512 pixels) were acquired by SICM at 25 °C in HBSS supplemented with 10 μM HEPES (Euroclone,

IT) to keep $CO_2$ concentration constant during the 'loop' images acquisition. Acquisition and analysis were performed with IonScope Ltd. UK software.

SEM imaging and Focused Ion Beam (FIB) sectioning were performed in a Zeiss Auriga Compact dual-beam microscope equipped with Schottky field emission gun, GEMINI column and Gallium ion gun. After 4 h culturing medium was removed and cells were rinsed in PBS (Sigma–Aldrich). Subsequently, cells were fixed in a 2.5% glutaraldehyde solution (Sigma–Aldrich) in Na-Cacodylate buffer (Sigma–Aldrich) for 30 min, washed in Na-Cacodylate buffer

**Fig. 7 Electrograms parameters and evaluation of epicardial velocities in the cryoinjury portion of the left ventricle following injection of SiC-NWs.**
**a** Top. Tissue excitability and refractoriness parameters were measured on the epicardial EGs. Pre: before cryoinjury, Post Cryo: after cryoinjury, Post Cryo + SiC-NWs: after cryoinjury and treatment with NWs. Activation time: Pre, $n = 66$; Post Cryo, $n = 58$; Post Cryo + SiC-NWs, $n = 28$. RT interval: Pre, $n = 63$; Post Cryo, $n = 56$; Post Cryo + SiC-NWs, $n = 35$. RR interval: Pre, $n = 45$; Post Cryo, $n = 54$; Post Cryo + SiC-NWs, $n = 26$. QRS complex: Pre, $n = 68$; Post Cryo, $n = 65$; Post Cryo + SiC-NWs, $n = 33$. **b** Representative color-coded isochrones activation maps obtained from vehicle-treated animals (top) and SiC-NWs treated animals (bottom) showing relief of conduction block (Pink arrow). $n = 4$ experiments repeated with similar results for Vehicle injection. $n = 8$ experiments repeated with similar results for SiC-NWs injection. **c** Phase-map analysis of instantaneous velocity in the selected electrodes for the three conditions: Pre, before cryoinjury ($n = 48$); Post Cryo, after cryoinjury ($n = 47$); Post Cryo + SiC-NWs, after cryoinjury and treatment with NWs ($n = 22$). Data expressed as median and interquartile range. *$p < 0.05$; **$p < 0.01$, ***$p < 0.001$ (Kruskal–Wallis; post-hoc analysis: Dunn's multiple comparison. C.I. = 95%). Source data are provided as a Source Data file.

(Sigma–Aldrich) for 5 min, and dehydrated in ethanol (Sigma–Aldrich) at increasing concentrations. MNBs were cut by FIB and cross-sections have been analyzed. The HL1 cells were seeded on a 1.9 cm² area coverslip. After 24 h, 50 μg/ml of SiC-NWs were added to the culture media. Finally, samples were sputtered with a thin layer of gold through an SCD 040 coating device (Balzer Union, Wallruf, DE). The SEM analysis was performed at 5 keV, while the cross-sectional analysis of MNB with FIB was performed with a Gallium ion beam at 30 kV with a current of 500 mA.

We determined the NW structure, size, crystallinity, and the capability to form a network in MI specimens by Transmission Electron Microscopy (TEM) analyses, performed on single NWs on a Field–Emission JEOL JEM-2200FS TEM microscope (JEOL Italia, Basiglio, Milano, IT) operated at 200 kV, either in conventional or scanning (STEM) mode. After the sacrifice, infarcted areas from left ventricles were extracted and fixed in 2.5% glutaraldehyde / 0.1 M PBS buffer at pH 7.2 for 1 h and then dehydrated through the graded series of acetone and embedded in Durcupan (Fluka Chemie, Buchs, CH). Polymerization occurred after 24 h at 65 °C. Sections of 2 μm were prepared using Reichert ultramicrotome (PabischWien, Leica Microsystem DE), colored with Toluidine blue 0.5% sodium carbonate, and observed under a light microscope. Ultrathin sections (~70 nm) were cut with a diamond blade, gathered on slotted copper grids, stained with uranyl acetate replacement stain (Electron Microscopy Science, PA, USA), and lead citrate. The micrographs were analyzed by ImageJ- Fiji software 1.51a.

We assessed the biodynamic interface of SiC-NWs using real-time live imaging of SiC-NW internalization in HL1 and primary cardiac myofibroblasts, with a Leica SP8 Laser-Scanning Confocal Microscope at ×63 with a NA 1.20 oil immersion objective (Leica). All experiments were carried out at 37 °C and 5% CO₂, using an incubation chamber enclosing the microscope stage and body. SiC-NWs were functionalized via Abberior 580-star dye (Abberior, DE), and cells were live-stained with CellMas Green Actin Tracking Stain (Thermofisher, IT). Analysis of colocalized actin on SiC-NWs was performed with Imaris 7.4.2 (Bitplane AG, CH).

The SiC-NWs treated cryoinjured myocardial tissue was dissected and fixed with 4% paraformaldehyde for 24 h before being immersed in 70% ethanol. Transmural MI w or w/o SiC-NWs were immediately acquired with a stereomicroscope (Leica System, USA) at ×20 and ×40.

**Junctional conductance in biological and synthetically coupled HL1 cell pairs.**
Dual whole-cell recordings were obtained with a Multiclamp 700B dual patch-clamp amplifier (Molecular Devices, Crisel Instruments, IT) controlled by pClamp software. Low-density cell cultures grown on glass coverslip were transferred to a custom-made experimental chamber that was fixed to the stage of an inverted microscope (Nikon eclipse, Ti/U) and superfused at 1.5 ml/min. Signals were filtered (3 kHz), digitized (50 kHz), and stored for offline analysis with dedicated software (Clampfit 10.6; Molecular Devices, Crisel Instruments, IT). Patch pipettes were pulled from borosilicate glass capillaries (Intrafil-10, INTRACEL LTD, UK) with a laser-based micropipette puller (P-2000; Sutter Instrument, USA) and resistances ranging from 2 to 4 MΩ. After seal formation (2-10 GΩ) and rupturing of the patch, liquid junction potentials (2 mV as calculated by pClamp software) were compensated. Adequate voltages control was achieved with membrane capacitances and series resistances ($r_s$: 3–8 MΩ) compensations above 80% (estimated voltage error <5 mV in all cases). Coupling conductance ($G_j$) of cell pairs was assessed using an established dual whole-cell patch-clamp protocols[55]. Experiments were performed on "biological" HL1-HL1 cell pairs coupled by connexins and "synthetically" HL1-HL1 cell pairs coupled by SiC-NWs or SiO₂-NWs. Preparations were superfused with an external solution containing (in mmol L⁻¹): NaCl 140, TEA-Cl 5.4, CaCl₂ 2, MgCl₂ 1, D-Glucose 1 and HEPES 10 (pH 7.40) and pipettes filling solution containing (in mmol L⁻¹): CsCl₂ 120, NaCl 10, MgATP 3, CsOH 16, EGTA 10, HEPES 5 (pH 7.2). Patch pipettes were mounted on separate micromanipulators (PatchStar Micromanipulator, Scientifica). After establishing successful whole-cell recording conditions, the membrane potential of both cells was clamped to −42 mV to inactivate sodium channels. While keeping the voltage of cell 1 constant, cell 2 was stepped to ±50 mV in 10 mV increments lasting 200 ms. Values for the resulting transjunctional currents (Ij) in cell 2 were determined at a steady state (Ij,ss). The protocol was then reversed (stepping of cell 1 recording of Ij,ss in cell 2). Thus, the $G_j$ for a cell pair was calculated as the average of the $G_j$ values measured in each direction. Furthermore, the absence of

significant differences between Ij,ss-1, and Ij,ss-2 indicated that Gj was not affected by asymmetric leak currents. Because membrane resistances were high compared to $r_s$, nonjunctional membrane currents values were therefore negligible and, a simplified electrical circuit with three resistances in series (rs1, rcoupling, rs2) was used to calculate gap junctional resistance[50]. Data were obtained from more than three different cultures for each experimental condition.

**Optical recording of action potential and intracellular calcium transient propagation.** Cell-to-cell action potential synchronization was assessed optically in the HL1 cell line w or w/wo SiC-NWs. Cells were cultured onto 22 mm coverslips; the preparations were stained using a voltage-sensitive dye (135 μM Di-8-ANEPPS, Biotium, USA) for 4 min and mounted on a custom-made temperature-controlled chamber placed on the stage of an inverted microscope for epifluorescence (Nikon Eclipse Ti/U, IT). Spontaneous action potential was assessed optically using a fast-resolution CMOS camera (L-Ultima, Scimedia, USA) and recorded at ×20 magnification (Nikon S Fluor, N.A. 0.75 Nikon, IT) for 4 s at 1–5 kHz temporal resolution. Intracellular Ca²⁺ transient was assessed optically in the same type of preparations. The preparations were loaded with 5 μmol/L of Fluo-4 AM (Thermofisher Scientific, IT) for 20 min and placed in the incubator before being mounted on a custom-made temperature-controlled chamber. Intracellular Ca²⁺ transient was assessed optically using a fast-resolution CMOS camera (L-Ultima, Scimedia, USA) and recorded at ×20 magnification for 4 s at 1 kHz temporal resolution. Preparations were perfused continuously with Hanks Balanced Salts Solution (EuroClone, IT) at 36 °C. During acquisition, the preparations were illuminated using a mercury light source (Intensilight, Nikon, IT). We did not apply extracellular stimulation to avoid stimulation artifacts on the cell sink.

**Experimental animals.** The study population consisted of 15 Sprague Dawley rats of both sexes bred in our animal facility. Rats (8–10 months old, weighing 300–350 g) were anesthetized by intraperitoneal injection of a mixture of 40 mg/kg ketamine and 0.15 mg/kg medetomidine and ventilated at ~85 cycles per min. The heart was exposed through a median thoracotomy and suspended in a pericardial cradle under artificial ventilation (RoVent® Small Animal Ventilator, Kent Scientific, CT, USA). Body temperature was maintained constantly at 37 °C with heath lamp radiation and further doses of anesthetic were administered as needed during the experiment. Animals were sacrificed with a lethal injection of sodium pentothal.

Local cryoinjury (Cryo) was performed via liquid-nitrogen delivery following a well-known protocol[25]. Cryoinjury was achieved by applying a 2 mm²-tip cryoprobe for 60 s on the anterior left ventricular epicardial surface after opening the chest. Three animals were monitored for 5 h following cryoinjury without intervention, four animals underwent injection with the vehicle, and the rest of the animals underwent injection of vehicle plus 1 mg of SiC-NWs.

The NWs were first detached from the growth substrate by ultrasonication in ethanol, then dried in a controlled atmosphere and stored in 1.5 mg batches (1 batch per animal). SiC-NWs were resuspended by ultrasonication in 150 μl saline solution and the suspension was transferred to a syringe with a short 30 G needle for myocardial injection. The syringe content was distributed triangularly, in three consecutive injections (1 ml in total) in the border of the 2 mm² infarcted regions keeping the three injection sites equidistant.

**In situ measurement of epicardial electrical activity.** In situ mapping of epicardial potential[13] was performed after exposing the heart. A custom-made silver electrode array (11 × 11 electrodes, resulting in 250 μm resolution square mesh), was placed onto the epicardial surface and used to record unipolar electrograms during sinus rhythm and ventricular pacing. Sinus rhythm parameters, longitudinal, transversal, and local velocities were evaluated at 16 kHz temporal resolution (MUX, Crescent Electronics, USA). The electrode array was positioned to cover the ventricular cryoinjury region and the following independent measurements were carried out in seven different cardiac cycles: EG parameters (P wave duration, PQ interval, QRS interval, QRS amplitude, QT interval, QTc, T-wave duration, RT and RR intervals); threshold (the minimal current capable to elicit ventricular activation for a 1 ms duration pulse); conduction velocities and

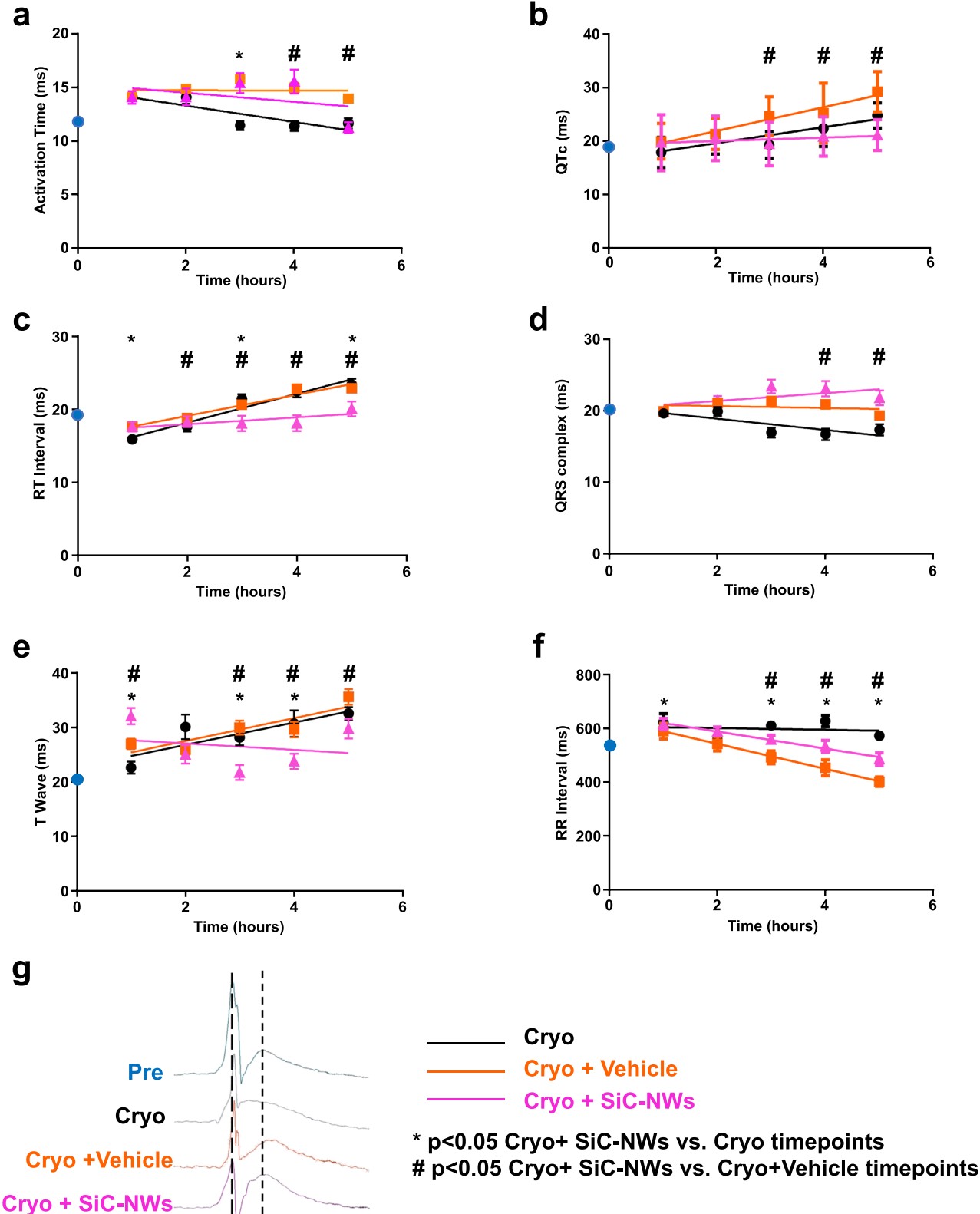

**Fig. 8 Recovery of epicardial electrograms parameters over time for the MI model used. a** Activation time. **b** QTc duration. **c** RT interval duration. **d** QRS complex duration. **e** T-wave duration. **f** RR interval duration. Blue dots: data measured before cryoinjury (Pre). Black dots: Cryo. Orange squares: Cryo + Vehicle. Pink triangles: Cryo + SiC-NWs. Colored lines: linear regression in the three conditions. Data are expressed as mean ± S.E.M. Statistical significance set at $p < 0.05$ (unpaired two-sided Student's $t$-test). **g** Time-matched EG overlapping of the four different conditions. Source data are provided as a Source Data file.

spatiotemporal phase maps[27] were evaluated independently in five electrodes for each animal.

**In situ measurement of cardiac kinematics**. Kinematic measurements of the beating rat hearts were performed using video kinematic evaluation of the epicardial movement[56]. Briefly, we used a computer-vision technology capable of acquiring videos of the epicardial motion during the sinus rhythm at 500 fps for 2 s (LabView). The acquisitions return kinematic parameters, such as cardiac force, cardiac energy, contractility, and ventricular mechanical compliance at the open chests (Pre Cryo), 2 h after Cryo and at the end of the experiments for the vehicle or SiC-NWs treatments.

**Statistics**. Data are expressed as mean ± SEM±, mean ± SD or as a median and interquartile range depending on their distribution. Normal distribution was checked by the Kolmogorov–Smirnov test. Statistics of variables included unpaired two-sided Student's *t*-test, Mann–Whitney U test, Kruskal–Wallis (post-hoc analyses: Dunn's multiple comparisons), and two-way ANOVA (post-hoc analyses: Bonferroni test or Games–Howell test, when appropriate). GraphPad 6.0 software (Prism, USA) was used to assess the normality of the data, the outliers and for the statistical calculation. The details of the specific test used for each experiment are reported in the figure legends. *P*-values of <0.05 were considered to be significantly different.

**Reporting summary**. Further information on research design is available in the Nature Research Reporting Summary linked to this article.

## Data availability

All data that support the findings described in this study are available within the manuscript. Preprocessed dataset regarding nonlinear microscopy, in vitro and in vivo electrophysiological and processed kinematic data are provided on repository website (https://doi.org/10.5281/zenodo.5711717). In vivo video preprocessed kinematic data are available upon reasonable request. Source data are provided with this paper.

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

## Acknowledgements

We acknowledge Giovanni Attolini, for the assistance in NWs batch preparation. We also acknowledge Serena Strozzi and Emilio Macchi for the helpful discussion regarding experimental procedures. This work has benefited from the equipment of the COMP-HUB Initiative, funded by the Departments of Excellence program of the Italian Ministry for Education, University and Research (MIUR, 2018-2022) to F.B. National Institute of Health (NIH, USA), Project #1R21CA223969-01A1, 2018-2021 to G.S. FIL_2018_Miragoli (Fondo Incentivante di Ateneo), Università di Parma to M.M. Horizon 2020 CUPIDO project GA: 720834 to D.C., M.M.

## Author contributions

M.M. and F.R. designed the project, P.L. and M.Q. performed nanomaterial fabrication. M.M., N.S., and P.L. performed in vitro electrophysiological experiments, R.A. and S.P. performed the toxicological investigation. J.M. and F.dA performed cell culture, confocal experiments, and analysis. S.R., G.R., F.P.L., and M.M. performed in vivo electrophysiological and kinematic experiments, F.G., P.L., and F.R. performed SEM and TEM analysis. M.M. and N.S performed patch-clamp and SICM acquisitions. F.B., G.S., D.C., and G.C. supervised the project. M.M. wrote the final version of the manuscript together with the contribution of all authors.

## Competing interests

The authors declare no competing interests.
