## [Peer Review File · Nature Communications]

REVIEWER COMMENTS

Reviewer #1 (Remarks to the Author):

As per an email from the editor, this reviewer focuses on the evaluation of the results and discussions of the nanowires studied in this work. The comments are as follows:

- (1) Can the authors expand the discussion of Figure 1D-1G? Researchers in other areas can benefit from the more in-depth discussion.
- (2) Can the authors provide more information about the nanowire electrical properties?
- (3) The internalization of nanowires by HL-1 cells or cardiac fibroblasts (they would be very relevant to in vivo studies) should be carefully evaluated. Can the authors perform real-time confocal microscope imaging to understand the biointerface dynamics in vitro?
- (4) The authors have used HPICM to study the topography of the HL-1 cells. Can HPICM provide any additional information on cell-cell electrical coupling?

Reviewer #2 (Remarks to the Author):

In this study, the investigators explore the use of a silicon-carbide nanowires to rapidly restore cardiac conduction post-myocardial infarction (MI). The authors provide evidence for biocompatibility of the nanowires, imaging data demonstrating membrane nanobridge formation and functional evidence of connections between myocytes with optical action potentials in vitro and epicardial electrode mapping in vivo. Overall this is a very innovative endeavor to address post-MI arrhythmias that furthers the group's prior work on electrically coupling myofibroblasts to cardiomyocytes. However, I would like to raise several issues:

1. Why were atrial rather than ventricular cardiomyocytes studied in culture?

2. From the data presented, the coupling of cells via membrane nanobridges seems weak with an average conductance of 3.98nS (per the manuscript text). Are the findings on significant effects on ECG parameters consistent with the weak overall conductance?

3. Can the authors explain why the action potential amplitude was decreased in the “sink cells” but not the intracellular Ca amplitude? The baseline for the calcium signals also appears very different in the “source” vs “sink” cells and on review of Figure 5A, the Ca²⁺ transient amplitude appears bigger in the sink cell (if baseline corrected).

4. With respect to the decrease in action potential amplitude, the authors mention electrotonic decay, but could this be expanded upon? The group’s prior work (references #33 and 34) do not seem to cover this topic.

5. It is difficult to interpret Figures 6C-D. Specifically, it is unclear what the yellow arrowheads in Figure 6C are showing. For Figure 6D, could more of the images be labeled?

6. While QRS amplitude is reported in supplemental Figure 3, are you able to comment on voltage mapping using the epicardial array during the in vivo studies?

7. For the statement: “increment of ...the RT interval (from 19.5±4.22 to 15.95±3.60 ms)”, did the RT interval increase or decrease with cryoinjury. Figure 8C seems to suggest an increase that then decreased with nanowires. Why is RT interval reported in Figures 7 and 8 but only QTc in Figure 8?

8. Can the authors explain why QRS duration increased after nanowire injection more so than in MI alone?

9. For Figure 8, are the p-values adjusted for multiple hypothesis testing? The statistics section mentions Dunn’s, but p-values are not reported as adjusted in Figure 8.

Minor comments:

1. Text is difficult to read in Figure 7.

2. Would rephrase “restoring ECGs” in abstract

3. This statement is confusing: "5-year cumulative incidence of SCD after coronary bypass graft (CABG) was 8.5% (between 31 to 90 day time period)". Did the authors intend to state that the highest rate during the 5-year period was during the 31-90 day time period?

4. Grammatical errors are found throughout the text.

Reviewer #1 (Remarks to the Author):

We thank the reviewer for his/her constructive comments that help to improve the manuscript

Q1: Can the authors expand the discussion of Figure 1D-1G? Researchers in other areas can benefit from the more in-depth discussion.

A1: We agree with the reviewer and we have expanded the discussion of Figure 1D-1G. We cited two more publications that described Silicon-based nanowires uptake (*Zimmermann et al.: Cellular uptake and dynamics of unlabeled freestanding silicon nanowires Sci Adv. 2016 Dec; 2(12): e1601039*) and the negligible effect on ROS production from SiC-NWs (*Chen et al.: Cellular toxicity of silicon carbide nanomaterials as a function of morphology. https://doi.org/10.1016/j.biomaterials.2018.06.027*). We also provide data regarding the internalization of SiC-NWs from HL1 and cardiac myofibroblasts, i.e. the cells that heavily populate myocardial infarction. The novel data also showed, as expected, partial internalization by the cells (see answer to Q3 and the new figure 3).

We have added the following sentence in the discussion section:

Page 22 Line 495 : SiC-NWs were not completely internalized by the cells⁴⁴ and, probably due to their structural properties, neither increase ROS production⁴⁵ nor affecting the cell cycle²³.

Q2: Can the authors provide more information about the nanowire electrical properties?

A2: A detailed analysis of the nanowires' electrical properties has been previously reported in our paper "Strain engineering of core-shell silicon carbide nanowires for mechanical and piezoresistive characterizations", by S. Nakata, A. Uesugi, K. Sugano, F. Rossi, G. Salviati, A. Lugstein and Y. Isono, published in *Nanotechnology* 30 (2019) 265702. Section 4.3 of this paper focuses on the I-V characteristics of single SiC- (Fig. 12b) and SiC/SiO₂-NWs contacted in a metal-semiconductor-metal device, in which the electrical transport is dominated by the semiconducting nanowire properties at a large bias. From the experimental data, SiC-NW was found to be n-type and its electrical conductivity was estimated to be 11.4 $\mu\text{S cm}^{-1}$.

We have added this reference (19) in the introduction section on

Page 3 Line 84...since they are chemically inert, semiconducting¹⁹ and compatible with the biological environment²⁰.

and have expanded the discussion section on

Page 22 Line 499. In our experiments, we compared the MNB formation as triggered by SiO₂- and SiC-NWs; only semiconducting SiC-NWs supported coupling, because of their conductivity (i.e. $\sim 11.4 \mu\text{S cm}^{-1}$)¹⁹, and therefore impulse propagation over distances.

Q3: The internalization of nanowires by HL-1 cells or cardiac fibroblasts (they would be very relevant to in vivo studies) should be carefully evaluated. Can the authors perform real-time confocal microscope imaging to understand the biointerface dynamics in vitro?

A3: We completely agree with the reviewer and we have performed the requested experiments. In detail, we have functionalized SiC-NWs with a specific 580 nm NHS fluorophore, and cell actin filaments have been labelled with green 488nm live dye. We performed the experiments via live confocal microscopy in HL1 cells and primary cardiac myofibroblasts as requested by the reviewer. We observed that after SiC-NW administration, actin covered $\sim 30\%$ of the SiC-NW length within 30 min, suggestive of rapid, dynamic (but partial) internalization (see new Figure 3, new Supplementary Figure 3 and new Supplementary Video 1 and Video 2).

We thank the reviewer for these suggestions, which certainly have improved our knowledge and the manuscript.

We made the following changes:

Methods:

Page 4 Line 107. For the confocal acquisitions, SiC-NWs have been functionalized with amino groups by reacting with 3-aminopropyltriethoxysilane (APTES) (2 μL of a solution 5 mM in toluene) in refluxing toluene. After washing with toluene and ethanol, the SiC-NWs were detached using a sonicating bath. The NWs were collected from ethanol by ultracentrifugation at 14000 Hz at 4 °C for 30 min and dried in the air. The NWs (1 mg) were dispersed in DMSO (1 μL) and 1 μL of DIPEA was added under magnetic stirring. Then, 200 μL of an NHS-Abberior STAR 580 solution in DMSO (1 mg/mL) were added. The reaction was stirred for 2 h at r.t. to allow the formation of the amide bond. Centrifugation at 14000 rpm at 4 °C, followed by three washing with DMSO, allowed to recover the NWs conjugated with the dye.

Page 4 Line 123. Neonatal rat ventricular myofibroblasts have been isolated and cultured as described elsewhere²⁸. Ethics permission was obtained from the Italian Ministry of Health (Protocol N. D2326/2019), and all isolations followed the directives of European law 63/2010 and Italian law 26/2014 for experimental animal use.

Page 6 Line 186.

Confocal Microscopy

We assessed the biodynamic interface of SiC-NWs using real-time live imaging of SiC-NW internalization in HL1 and primary cardiac myofibroblasts, with a Leica SP8 Laser-Scanning Confocal Microscope at 63X with a NA 1.20 oil immersion objective (Leica). All experiments were carried out at 37 °C and 5% CO₂, using an incubation chamber enclosing the microscope stage and body. SiC-NWs were functionalized via Abberior 580-star dye (Abberior, DE), and cells were live-stained with CellMas Green Actin Tracking Stain (Thermofisher, IT). Analysis of co-localized actin on SiC-NWs was performed with Imaris 7.4.2 (Bitplane AG, CH).

Results:

Page 13 Line 350 ..., more than a single SiC-NW can be covered by the same MNB, resulting in the capability to connect more cells creating a synthetic network of cell activity in vitro. Live-confocal imaging revealed that MNB formation was orchestrated by actin filaments that linearly co-localized with the SiC-NWs by ~30% after 30 min in HL1 cells (Fig. 3b,c, Supplementary Video 1) and in cardiac myofibroblasts (Supplementary Fig. 3, Supplementary Video 2).

Discussion:

Page 22 Line 492 . In the current study, we grew SiC-NWs which do not require cell grafting for the support of impulse propagation but, once cultured with cardiomyocytes or myofibroblasts, can create actin-triggered MNB networks that synthetically transfer the electrotonic potential from cell-to-cell over distance.

Conclusions

Page 23 Line 533 ...biophysically inspired SiC-NWs to create actin-based MNB networks, mimicking the physiological cell-cell cardiac communication.

Q4: The authors have used HPICM to study the topography of the HL-1 cells. Can HPICM provide any additional information on cell-cell electrical coupling?

A4: We have added more data regarding cell-cell coupling. As expected, the observed heterogeneity in junctional conductance for the “control” condition (14.13 ± 2.41 nS) depends not only on the gap junctional activities but also on the cell-cell contact length (9.36 ± 6.88 μm). This is not the case when cells are synthetically coupled by SiC-NWs (3.98 ± 0.97 μm), as the contact site depends on the MNB formed onto the SiC-NW (4.70 ± 0.6 μm). By means of HPICM, we measured the contact site of the coupled cells in both conditions, and new data are reported in the new Figure 2. We thank the reviewer for this question, as the answer underlined the significant difference in conductance and contact length (new Fig. 2C).

We have made the following changes:

Result Section:

Page 11 Line 325 : This discrepancy was caused by the more heterogeneous lengths of the cell-cell contacts in the natural (9.36 ± 6.88 μm) vs. synthetic (4.70 ± 0.65 μm) condition. We obtained similar MNBs generation and thus the fusion with the distance cell by SiO₂-NWs (Supplementary Fig.2), but there was no junctional current (Supplementary Fig.2b), a finding suggesting that only semiconductive SiC-NWs and not insulated SiO₂-NWs are capable to support junctional current over distance.

Discussion Section:

Page 22 Line 502 : However, SiC-NWs coupled cells had reduced junctional currents compared to physically coupled HL1 cardiomyocytes⁴⁸, but this reduction does not impede the correct AP transfer via MNB formation.

Weingart et al. and Wilder et al. showed in in-vitro coupled cardiomyocytes and mathematical simulations that AP transfer is possible for electrical conductance values lower than 3.98 nS^{49, 50}. Nonetheless, the junctional current not only depends on the degree of coupling but also on the number of gap junctions present⁴². This is not the case when HL1 cells were synthetically coupled by SiC-NWs, wherein the contact lengths depended on MNB edge dimensions (cf. Fig 2c). Such

coupling allowed sustainable AP propagation (albeit reducing AP amplitude in sink cells, possibly on account of electrotonic impulse propagation delays across MNBs³⁵) and intracellular Ca²⁺ transient synchronization in the cell network.

Reviewer #2 (Remarks to the Author):

In this study, the investigators explore the use of silicon-carbide nanowires to rapidly restore cardiac conduction post-myocardial infarction (MI). The authors provide evidence for biocompatibility of the nanowires, imaging data demonstrating membrane nanobridge formation and functional evidence of connections between myocytes with optical action potentials in vitro and epicardial electrode mapping in vivo. Overall this is a very innovative endeavor to address post-MI arrhythmias that furthers the group's prior work on electrically coupling myofibroblasts to cardiomyocytes. However, I would like to raise several issues:

Q1. Why were atrial rather than ventricular cardiomyocytes studied in culture?

A1. We completely understand the reviewer's issue and we have tested the same SiC-NWs internalization in neonatal ventricular myocytes and neonatal myofibroblasts. However, due to the morphology acquired once in culture, neonatal ventricular myocytes tend to form spontaneous MNBs even in the absence of SiC-NWs (see figure below).

Therefore, it is not possible from our side to discriminate biological from synthetic aetiology of MNBs, especially during optical mapping and double patch clamp experiments (equipped with a classic inverted fluorescence microscope).

We also tested SiC-NWs on seeded adult ventricular myocytes. However, adult CMs do not resist in culture for a long time, and we were not technically capable of observing in vitro SiC-NW internalization.

We thus selectively focused on the HL-1 atrial cell line because of their round morphology once cultured (and the production of MNBs in the presence of SiC-NWs), as well as their electromechanical activity resolvable with HPICM, optical mapping and double patch clamp.

To overcome this concern, we have added more data regarding the biodynamic interface of labelled SiC-NWs in HL1 cells and primary neonatal cardiac ventricular myofibroblasts (see novel Figure 3, Supplementary Figure 3, Supplementary Video 1 and 2) using real-time confocal microscopy, hoping that this answer satisfies the reviewer's concern.

We have made the following changes:

Methods:

Page 4 Line 107. For the confocal acquisitions, SiC-NWs have been functionalized with amino groups by reacting with 3-aminopropyltriethoxysilane (APTES) (2 μ L of a solution 5 mM in toluene) in refluxing toluene. After washing with toluene and ethanol, the SiC-NWs were detached using a sonicating bath. The NWs were collected from ethanol by ultracentrifugation at 14000 Hz at 4 °C for 30 min and dried in the air. The NWs (1 mg) were dispersed in DMSO (1 μ L) and 1 μ L of DIPEA was added under magnetic stirring. Then, 200 μ L of an NHS-Abberior STAR 580 solution in DMSO (1 mg/mL) were added. The reaction was stirred for 2 h at r.t. to allow the formation of the amide bond. Centrifugation at 14000 rpm at 4 °C, followed by three washing with DMSO, allowed to recover the NWs conjugated with the dye.

Page 4 Line 123. Neonatal rat ventricular myofibroblasts have been isolated and cultured as described elsewhere²⁸. Ethics permission was obtained from the Italian Ministry of Health (Protocol N. D2326/2019), and all isolations followed the directives of European law 63/2010 and Italian law 26/2014 for experimental animal use.

Page 6 Line 186.

Confocal Microscopy

We assessed the biodynamic interface of SiC-NWs using real-time live imaging of SiC-NW internalization in HL1 and primary cardiac myofibroblasts, with a Leica SP8 Laser-Scanning Confocal Microscope at 63X with a NA 1.20 oil immersion objective (Leica). All experiments were carried out at 37 °C and 5% CO₂, using an incubation chamber enclosing the microscope stage and body. SiC-NWs were functionalized via Abberior 580-star dye (Abberior, DE), and cells were live-stained with CellMas Green Actin Tracking Stain (Thermofisher, IT). Analysis of co-localized actin on SiC-NWs was performed with Imaris 7.4.2 (Bitplane AG, CH).

Results:

Page 13 Line 350 ..., more than a single SiC-NW can be covered by the same MNB, resulting in the capability to connect more cells creating a synthetic network of cell activity in vitro. Live-confocal imaging revealed that MNB formation was orchestrated by actin filaments that linearly co-localized with the SiC-NWs by ~30% after 30 min in HL1 cells (Fig. 3b,c, Supplementary Video 1) and in cardiac myofibroblasts (Supplementary Fig. 3, Supplementary Video 2).

Discussion:

Page 22 Line 492 . In the current study, we grew SiC-NWs which do not require cell grafting for the support of impulse propagation but, once cultured with cardiomyocytes or myofibroblasts, can create actin-triggered MNB networks that synthetically transfer the electrotonic potential from cell-to-cell over distance.

Conclusions

Page 23 Line 533 ...biophysically inspired SiC-NWs to create actin-based MNB networks, mimicking the physiological cell-cell cardiac communication.

Q2. From the data presented, the coupling of cells via membrane nanobridges seems weak with an average conductance of 3.98nS (per the manuscript text). Are the findings on significant effects on ECG parameters consistent with the weak overall conductance?

A2. We thank the reviewer for this concern about the apparent weak average conductance mediated by MNB pairing of HL-1 cells. In the literature, it has been shown that electrical coupling of 3.98nS, as we measured via SiC-NWs synthetic coupling, is capable of sustaining action potential transfer in cell pairs obtained from adult rat and guinea pig ventricles (Weingart and Maurer: *Circulation Research*. 1988;63:72–80). In this paper, impulse transfer was completely blocked once the intercellular coupling resistance was greater than 780 MOh (~1.3nS). Also in Wilders' paper (Wilders et al.: *Action Potential Conduction Between a Ventricular Cell Model and an Isolated Ventricular Cell*. *Biophys J*. 1996 Jan; 70(1): 281–295), it was shown that action potential transfer between a ventricular cell model and an isolated ventricular cell is possible for electrical conductance values even lower than 4nS.

Furthermore, TEM images of the infarcted cardiac tissue (fig. 6, D) show that nanowires form a network of several connected portions of tissue. This condition is substantially different from the one occurring in vitro for double patch-clamp experiments, whereby cell preparations were made for the specific purpose of quantifying the electrical conductance of single HL-1 cells coupled by single MNBs.

For this purpose, we have added more data regarding cell-cell coupling. As expected, the observed heterogeneity in junctional conductance for the “control” condition (14.13 ± 2.41 nS) depends not only on the gap junctional activities but also on the cell-cell contact length (9.36 ± 6.88 μm). This is not the case when cells are synthetically coupled by SiC-NWs (3.98 ± 0.97), as the contact site depends on the MNB formed on the NW (4.70 ± 0.6 μm at the isthmus). By means of HPICM, we measured the contact site of the coupled cells in both conditions, and the new data have been given in the new Figure 2.

While the in-vitro data only estimate the coupling gradient between two cells, the reviewer correctly supposed the consistency with EG parameters and conductance calculated by spatiotemporal phase mapping. The EG parameters presented in this work are a local average of the electrical activity for the entire heart. Even if SiC-NWs were injected only in a small part of the heart ($8\text{-}10$ mm^2), we did not observe a significant increase in QRS duration (see A6 and A8 below).

We have added the following sentences within the manuscript:

Result Section:

Page 11 Line 325: This discrepancy was caused by the more heterogeneous lengths of the cell-cell contacts in the natural (9.36 ± 6.88 μm) vs. synthetic (4.70 ± 0.65 μm) condition. We obtained similar MNBs generation and thus the fusion with the distance cell by SiO₂-NWs (Supplementary Fig.2), but there was no junctional current (Supplementary Fig.2b), a finding suggesting that only semiconductive SiC-NWs and not insulated SiO₂-NWs are capable to support junctional current over distance.

Discussion Section:

Page 22 Line 502: However, SiC-NWs coupled cells had reduced junctional currents compared to physically coupled HL1 cardiomyocytes⁴⁸, but this reduction does not impede the correct AP transfer via MNB formation.

Weingart et al. and Wilder et al. showed in in-vitro coupled cardiomyocytes and mathematical simulations that AP transfer is possible for electrical conductance values lower than 3.98 nS^{49, 50}. Nonetheless, the junctional current not only depends on the degree of coupling but also on the number of gap junctions present⁴². This is not the case when HL1 cells were synthetically coupled by SiC-NWs, wherein the contact lengths depended on MNB edge dimensions (cf. Fig 2c). Such coupling allowed sustainable AP propagation (albeit reducing AP amplitude in sink cells, possibly on account of electrotonic impulse propagation delays across MNBs³⁵) and intracellular Ca²⁺ transient synchronization in the cell network.

Q3. Can the authors explain why the action potential amplitude was decreased in the “sink cells” but not the intracellular Ca amplitude? The baseline for the calcium signals also appears very different in the “source” vs “sink” cells and on review of Figure 5A, the Ca²⁺ transient amplitude appears bigger in the sink cell (if baseline corrected).

A3. The AP amplitude in the sink cells is decreased (significantly vs CTRL cells and not significantly vs source cells), while this was not the case for intracellular Ca²⁺ because two different mechanisms are at the basis of their generation. In the first case, the generation of the action potential in the sink cell is strictly linked to the capability of the excitatory current generated by the source cell to rapidly depolarize the sink cell membrane to Na⁺ channel activation threshold. The transfer of the impulse from the source cell to the sink cell through the MNB is delayed based on the resistive properties of the MNB itself. The impulse propagation delay across the MNB causes a slower passive depolarization of the sink cell membrane with respect to the source cell, determining inactivation of Na⁺ channels, which in turn causes a delay in the onset of the AP and a decrease in I_{Na} in the sink cell.

The intracellular Ca²⁺ amplitude is determined by a different mechanism not related to the depolarization of the cell membrane and the electrical conduction properties of MNBs, but that relies on the complex calcium-induced calcium release (CICR) mechanism, which describes the biological process by which calcium can activate the release of calcium from the sarcoplasmic reticulum.

The baseline for the calcium traces is now corrected. Thank you.

Q4. With respect to the decrease in action potential amplitude, the authors mention electrotonic decay, but could this be expanded upon? The group’s prior work (references #33 and 34) do not seem to cover this topic.

A4. We thank the reviewer for his/her constructive comment that has allowed us to better explain the mechanism that generates the AP amplitude decrease in the sink cell.

As explained above, impulse transmission delay from the source cell to the distal sink cell across the MNB generates a partial inactivation of Na⁺ channels due to slow subthreshold charging during the prolonged foot potential (Rudy and Wang: Action potential propagation in inhomogeneous cardiac tissue: safety factor considerations and ionic mechanism. *Am J Physiol Heart Circ Physiol.* 278: H1019–H1029, 2000). This mechanism is similar to what has been previously described by Gaudesius et al. in heterocellular cultures consisting of cardiomyocyte strands interrupted by cardiac fibroblasts over defined distances (Reference 33.). Cardiac fibroblasts or myofibroblasts (MFBs) do not express Na⁺ channels or L-type Ca²⁺ channels, and the only sustained impulse propagation is the electrotonus (the MFB membrane behaves like an ohmic conductor). APs in the sink cells were delayed in a length-dependent fashion, and similarly decreased in amplitude as we

measured by connecting HL-1 cells through MNBs. Impulse propagation along MFB inserts was sustained over distances up to $\sim 350 \mu\text{m}$, and optical action potential and intracellular Ca^{2+} synchronization along MNBs were possible up to approximately $350 \mu\text{m}$. References 33 and 34 have been cited to explain the electronic impulse propagation delay mediated by MNBs and source and sink HL-1 optical APs characteristics.

It is more correct to rephrase the sentence in the discussion section writing that the decrease in sink cell action potential amplitude is due to impulse propagation delay rather than electrotonic decay.

We have made the following changes:

Discussion Section:

Page 22 Line 509: Such coupling allowed sustainable AP propagation (albeit reducing AP amplitude in sink cells, possibly on account of electrotonic impulse propagation delays across MNBs³⁵) and intracellular Ca^{2+} transient synchronization in the cell network.

Q5. It is difficult to interpret Figures 6C-D. Specifically, it is unclear what the yellow arrowheads in Figure 6C are showing. For Figure 6D, could more of the images be labeled?

A5. In the new Fig. 6C, the left panel represents coring of the cryoinjured area, in which is visible a whitish region damaged by the inserted needle, and a grey region containing the SiC-NWs. The right panel represents a zoomed region that highlights in grey the presence of SiC-NWs. The yellow arrowheads indicate the presence of the SiC-NW network after the injection. Figure 6D is now changed and labelled, while the legend has been modified accordingly.

Q6. While QRS amplitude is reported in supplemental Figure 3, are you able to comment on voltage mapping using the epicardial array during the in vivo studies?

A6. We are sorry for this mistake, but all the parameters, including QRS amplitude, were evaluated on the electrograms (EGs) recorded on the epicardial surface (on the RMS signal) and not on the surface ECG. We monitored several EG parameters, including QRS amplitude, as indicators for the cryoinjury approach. As expected, QRS amplitude decrease after cryoinjury, but after 2 hrs it became stable (see Supplementary Fig. 3).

We have added the following sentence on the

Page 17 Line 417...; indeed, QRS amplitude – an indicator of damaged tissue – decreased immediately after cryoinjury, and stabilized after 2 h.

Q7. For the statement: “increment of ...the RT interval (from 19.5 ± 4.22 to 15.95 ± 3.60 ms)”, did the RT interval increase or decrease with cryoinjury. Figure 8C seems to suggest an increase that then decreased with nanowires. Why is RT interval reported in Figures 7 and 8 but only QTc in Figure 8?

A7. We apologize for the mistake. The RT interval as you highlighted decreases after cryoinjury. We have corrected the sentence accordingly.

In Fig.8C we reported longitudinal changes of EG parameters over time. Still evident is a decrement in the RT after cryoinjury (blue dot – control – vs second black dot – Cryo 2 hrs) and a recovery of the RT value after SiC-NWs injection (blue dot – control – vs last green dot – SiC-NW treatment).

We report in Figure 7 the parameters that were significantly changed only for the group that underwent SiC-NW treatment, while in Fig. 8 we report the longitudinal EG parameters over time for three groups, i.e. Cryo, Cryo+Vehicle, Cryo+SiC-NWs.

Q8. Can the authors explain why QRS duration increased after nanowire injection more so than in MI alone?

A8. We understand the reviewer's point as QRS in Figure 8 (green line) seems to increase over time. The statistical approach used here (unpaired Student t-test) was used to highlight only the significant difference for each time-point, therefore we did not include, on purpose, the statistical analysis over time. Following the reviewer's suggestions, we have re-analyzed the QRS data from 1 to 5 hrs after injection, and a Kruskal-Wallis tests (post hoc analyses: Dunn's multiple comparisons) did not show significance over time. The Vehicle data displayed the same behaviour over time (no significance). Similarly, the observed increment in the presence of SiC-NWs (from 19.96 ± 4.45 (1hr after injection) to 21.85 ± 6.05 (5 hrs after injection)) is negligible. If the reviewer intended the difference between MI alone and SiC-NWs treatment in Fig.7, the statistical test was, as well, not significant.

Q9. For Figure 8, are the p-values adjusted for multiple hypothesis testing? The statistics section mentions Dunn's, but p-values are not reported as adjusted in Figure 8.

A9. As mentioned above, in Figure 8 we performed statistical analysis only for each time point and not over time. We used an unpaired Student's t-test to highlight the differences between groups in each time-point. We agree with the reviewer and we have clarified this in each figure legend for the applied tests.

Minor comments:

1. Text is difficult to read in Figure 7.

The figure is now redrawn with increased font

2. Would rephrase “restoring ECGs” in abstract

We have rephrase the sentence in the abstract as follows:

allow rapid reinstatement of impulse propagation across damaged areas and recover electrogram parameters and conduction velocity

3. This statement is confusing: “5-year cumulative incidence of SCD after coronary bypass graft (CABG) was 8.5% (between 31 to 90 day time period)”. Did the authors intend to state that the highest rate during the 5-year period was during the 31-90 day time period?

Yes, sorry about that. The numerically greatest monthly rate of SCD was in the 31- to 90-day time period.

We have change the sentence as follows:

...the 5-year cumulative incidence of SCD after coronary bypass graft (CABG) was 8.5% (the highest monthly rate of SCD occurred in the 31- to 90-day time period)²,....

4. Grammatical errors are found throughout the text.

We have corrected the grammatical errors and the English throughout the manuscript.

REVIEWERS' COMMENTS

Reviewer #1 (Remarks to the Author):

The authors have addressed all my concerns. The publication is recommended.

Reviewer #2 (Remarks to the Author):

The revision is very responsive to the comments. I have no further suggestions and look forward to the replication/future development of this novel approach.